# UNSW Face Test: A screening tool for super-recognizers

**James D. Dunn**[1]*, **Stephanie Summersby**[1], **Alice Towler**[1], **Josh P. Davis**[2], **David White**[1]

**1** School of Psychology, UNSW Sydney, Kensington, NSW, Australia, **2** Department of Psychology, University of Greenwich, London, United Kingdom

* j.d.dunn@unsw.edu.au

## Abstract

We present a new test–the UNSW Face Test (www.unswfacetest.com)–that has been specifically designed to screen for super-recognizers in large online cohorts and is available free for scientific use. Super-recognizers are people that demonstrate sustained performance in the very top percentiles in tests of face identification ability. Because they represent a small proportion of the population, screening large online cohorts is an important step in their initial recruitment, before confirmatory testing via standardized measures and more detailed cognitive testing. We provide normative data on the UNSW Face Test from 3 cohorts tested via the internet (combined n = 23,902) and 2 cohorts tested in our lab (combined n = 182). The UNSW Face Test: (i) captures both identification memory and perceptual matching, as confirmed by correlations with existing tests of these abilities; (ii) captures *face-specific* perceptual and memorial abilities, as confirmed by non-significant correlations with non-face object processing tasks; (iii) enables researchers to apply stricter selection criteria than other available tests, which boosts the average accuracy of the individuals selected in subsequent testing. Together, these properties make the test uniquely suited to screening for super-recognizers in large online cohorts.

## Introduction

People show a surprising degree of variation in their ability to identify faces, ranging from chance-level to perfect accuracy. These individual differences are stable over repeated testing (e.g. [1]), generalise from one face identification task to another (e.g. [2]), and represent a domain-specific cognitive skill that is dissociable from general intelligence (e.g. [3]) and visual object processing ability (e.g. [4]; c.f. [5]). Moreover, twin studies reveal this ability is highly heritable [6, 7]. Together, this evidence indicates that face identification ability is a stable cognitive trait with a biological basis, which means it can be reliably measured.

Face identification ability is normally distributed, and people at the very top end–'super-recognizers'–demonstrate extraordinary abilities [8, 9]. Therefore, super-recognizers could make substantial contributions to the theoretical understanding of face identification, by helping to pinpoint the cognitive, perceptual, and neural mechanisms underlying accurate

**Data Availability Statement:** All datafiles are available from the OSF database (DOI:10.17605/OSF.IO/E4TYG).

**Funding:** This research was supported by an Australian Research Council Linkage grant to DW

(LP160101523; arc.gov.au/linkage-projects) in
partnership with the Department of Foreign Affairs
and Trade, Australian Passport Office, an Australian
Research Council Discovery grant to DW
(DP190100957; arc.gov.au/grants/discovery-
program), and a UNSW Scientia Fellowship
awarded to DW (scientia.unsw.edu.au/scientia-
fellowships). The funders had no role in study
design, data collection and analysis, decision to
publish, or preparation of the manuscript.

**Competing interests:** The authors have declared
that no competing interests exist.

identification [10–12]. They can also make important practical contributions by working in applied face identification roles to reduce error-rates in law enforcement [13], criminal trials [14, 15], and security-critical identity management tasks ([1, 16, 17]; see also [18]).

However, *finding* super-recognizers is challenging because they make up just 2–3% of the general population. Initial work in this area recruited super-recognisers via self-report measures and anecdotal claims of superior ability in response to participant recruitment adverts (e.g. [9, 19]). But identifying super-recognizers based on self-report alone is unreliable [20, 21], with high performers consistently underestimating their own face identification ability [22]. As a result, screening based on self-report is better suited for detecting deficits–rather than superiority–in face identification ability (e.g. PI-20; [23]).

A more promising method of finding super-recognizers is to administer cognitive tests of face identification ability and map people's performance to a normative population [20, 24, 25]. The two most commonly used tests are the Cambridge Face Memory Test (CFMT/CFMT +; [9, 26]), and the Glasgow Face Matching Test (GFMT, [27]). Both tests are calibrated psychometric tests intended to be reliable measures of a person's ability and used in academic research [20, 24, 25] and professional recruitment [13, 18, 28]. However, these tests are increasingly used as online screening tools, which may limit their ongoing use for psychometric testing. For example, if participants make repeated test attempts online, it could lead to inflated estimates of ability. This may be especially true for those with superior face memory ability, who are likely to benefit more from repeated exposure to faces in memory tasks than the average person [29]. To preserve the function of these psychometric measures as diagnostic, confirmatory tests of face identification ability, new online screening tests are required.

Here we present the UNSW Face Test, an online screening tool for super-recognizers. This test has been designed specifically to reliably identify super-recognizers in large scale online testing. The UNSW Face Test is *not* designed to replace existing tests, but to complement them by providing a screening tool that can identify individuals that are likely to excel on existing tests. Three main properties of the UNSW Face Test distinguish it from existing tests and make it ideally suited to identifying super-recognizers. First, given the sole aim of the test is to identify super-recognisers, we did not calibrate the difficulty of the test so that mean accuracy was centred on the midpoint of the measurement scale, as is common practice in standardised psychometric tests [30]. Instead, we designed the task to be very challenging, such that average performers performed at the lower end of the scale. Super-recognizers typically achieve ceiling or near-ceiling accuracy on existing standardised tests (for example see [9, 11, 13]). By shifting the mean towards the lower end of the scale we enabled more precise stratification of abilities in the upper tail of the test score distribution. This allows stricter thresholds to be applied at the screening phase so that even if someone scores lower on subsequent confirmatory tests because of regression to the mean effects, they are unlikely to drop so far that they will not meet super-recognizer criteria on those tests.

Second, we designed the UNSW Face Test so that it captures people with a *general* ability to identify faces, across memory and matching tasks. The CFMT is designed to test face recognition memory, and the GFMT is designed to test face matching ability. These abilities are employed to greater or lesser degrees in different professional tasks that super-recognizers have been recruited to perform. For example, in CCTV surveillance, super-recognizers might monitor footage for faces they have committed to memory (e.g. [31]), whereas passport officers typically match photo-ID to unfamiliar travellers. While these abilities may be dissociable to a limited extent (e.g. [2, 16, 25, 32]), the high correlation between them suggests there is substantial overlap in these two abilities [11, 33, 34]. Identifying people with a general face identification ability, therefore, allows researchers to follow up initial screening with more detailed

profiling of participants' abilities, tailored to the specific identification task of interest (see [18]).

Third, images in the UNSW Face Test capture natural 'ambient' variability in appearance—caused by changes in age, pose, lighting and expression. The GFMT and CFMT use highly standardised images captured under optimal studio conditions in the same session. These types of images do not reflect the challenge of real-world face identification (see [35]). Using ambient images to test face identification ability, therefore, more closely approximates real-world face identification tasks, and makes the task challenging without having to resort to artificial degradation of test images, such as in the CFMT+ (for example see [25]). In this paper, we describe the development and validation of the UNSW Face Test. We find that the UNSW Face Test is a valid and reliable test that is uniquely suited to screening for super-recognizers.

## Test development

### Test delivery

The UNSW Face Test is free for use in research and can be completed at www.unswfacetest.com. Unique weblinks can be created for researchers and organizations to perform their own screening. Those interested in using the test should complete the following web form (http://www.unswfacetest.com/request.html). A package containing jsPsych functions [36], experiment scripts and images can also be provided on request. This enables researchers to create their own versions of the test using image datasets collected using the protocol described below. This may be desirable if researchers wish to target super-recognizers in a particular demographic group (for example see S1 Appendix for analysis of accuracy by participant ethnicity; see also [37]).

### Image selection and preparation

Images in the UNSW Face Test were selected from a database of 236 consenting undergraduate students at UNSW Sydney (for full details see [38]). From this database, we selected 40 'target' identities and a similar-looking foil identity. The mean age of the faces was 19 years old ($SD = 1.4$) and there was an equal number of males and females. Target-foil pairs were selected collaboratively by the research team whose goal was to find the individuals that were the most similar in facial appearance to each other in the database. Because we were designing a test to identify super-recognizers in Australia, we chose faces to be heterogeneous with respect to ethnicity and that reflect the diversity of the Australian population. Half the individuals self-identified as being of European descent. The other half self-identified as being as either Asian descent (5 were used as targets in the Memory task, 10 in the Sort task) or of Indian descent (5 as targets the Memory task). We then selected 5 images of each target and foil face. One was a studio-quality, frontal SLR image taken in controlled lighting and capture conditions with a neutral expression. The remaining four were ambient images subjects provided from their social media accounts, with uncontrolled variation in head angle, expression, lighting conditions, image quality etc. This led to some differences in the native resolution of images in these sets, but we used images in the test that had a minimum of 60 pixels between the eyes. Each image was rotated so the eyes were level, cropped to a 2:3 ratio and then resized to be 400 pixels wide at a resolution of 72 pixels per inch.

### Test design and procedure

The UNSW Face Test consists of two tasks completed in a fixed order: a recognition memory task and a match-to-sample sorting task (see Fig 1). A participant's score on the UNSW Face

# Recognition Memory Task

# Match-to-Sample Sorting Task

**Fig 1. UNSW face test design.** The UNSW Face Test contains two tasks. Left: In the recognition memory task participants study studio-quality target faces for 5 seconds each (Study Phase), and then make old/new recognition judgments on ambient test faces (Test Phase). Right: In the match-to-sample sorting task participants memorize a studio-quality target face for 5 seconds and then sort 4 ambient test images according to whether they are the target face. Scores on each task are summed for a maximum score of 120 and then expressed as a percentage.

Test is the sum of both tasks, for a maximum score of 120, which is then expressed as percentage accuracy. For analysis of results separately by sub-task see S2 Appendix.

**Recognition memory task.** In this task, participants complete a standard old/new recognition memory paradigm. In the study phase, participants memorize 20 studio-quality target faces, shown for 5 seconds each in random order. In the test phase, participants see 20 ambient images of the targets randomly intermixed with 20 ambient foil faces and decide whether each face appeared or did not appear in the study phase. Participants make 40 decisions, giving participants a maximum score of 40. This task takes approximately 5 minutes.

**Match-to-sample sorting task.** This task combines immediate face memory, perceptual matching and pile sorting (see [35]), and is designed to model 'identity clustering', a common task in criminal investigations where police determine which of multiple images shows the person of interest. On each trial, participants memorise a studio-quality target face for 5 seconds. Next, they sort a 'pile' of four ambient images by dragging an image to the right if it shows the target or to the left if it does not. Participants are told the pile could contain between 0 and 4 images of the target. The remaining images in the set of 4 are of the target's foil. Participants complete two practice trials, followed by 20 trials in a fixed order. There is an equal number of trials (4) containing 0, 1, 2, 3, 4 target images. Participants' scores on the match-to-sample sorting task are calculated as the total number of correct target and non-target classifications, giving a maximum score of 80. This task takes approximately 8 minutes.

## Test validation approach

To validate the UNSW Face Test, we recruited a total of 24,084 participants across 5 samples. For brevity, we provide basic information about the participant groups and procedures below and report full details in the method section at the end of the paper. All data is available via OSF: https://osf.io/35npj/.

First, we established normative accuracy on the test using an online sample of 290 participants from Amazon's Mechanical Turk. This is currently the most popular source of online

participant recruitment and is substantially more diverse than undergraduate students in terms of age, demographics, and education level [39–42], making it suited to establishing initial normative performance on our test.

Next, because the UNSW Face Test is designed to identify super-recognizers within large online cohorts, we recruited two large online samples that were recruited via online advertisements and news articles about super-recognizers. Online Sample 1 consists of 22,776 people who completed the UNSW Face Test via www.unswfacetest.com between September 7, 2017, and August 23, 2019, after following links in various news media (e.g. [43]). Online Sample 2 consists of 836 people who completed the UNSW Face Test, CFMT+ and GFMT between March 7, 2018, and August 1, 2019, after responding to an online advertisement by The University of Greenwich. We used these online samples to (i) establish normative accuracy on the UNSW Face Test for online samples resulting from super-recognizer focused recruitment, (ii) confirm the test is sufficiently difficult to avoid ceiling effects in these cohorts, (iii) compare the test's effectiveness as a screening tool to existing tests, and (iv) examine the effect of participant demographics on accuracy.

Finally, we recruited two lab-based samples of university students from UNSW to establish the fundamental psychometric properties of the test. Lab Sample 1 consists of 80 participants who completed the UNSW Face Test, GFMT and CFMT, and then returned one week later to repeat the UNSW Face Test. We used this sample to assess (i) test-retest reliability, which is important to confirm the test will be a useful screening tool, and (ii) convergent validity, which confirms the test measures skill in face identification. Lab Sample 2 consists of 102 participants who completed the UNSW Face Test, CFMT+, and non-face tasks: the Cambridge Car Memory Test [44], and Matching Familiar Figures Test [45]. We used this sample to assess discriminant validity, which confirms the test measures face identification skills rather than domain-general object processing skills. Participants in both lab samples also completed an ethnicity questionnaire.

## Performance on the UNSW Face Test

### Normative accuracy

We established normative performance on the UNSW Face Test using 290 participants from Amazon's Mechanical Turk (see Method for full details). Mean normative accuracy was 58.9% (SD = 5.8%) and participants responses ranged from 45.0% to 75.0%. The standard super-recognizer criterion is 2 SDs above the mean, which would make the super-recognizer threshold 70.5% on the UNSW Face Test (but see below for the benefits of using a stricter threshold). For brevity, analysis of accuracy and response time by sub-test is reported in S2 Appendix.

Fig 2 shows the normative distribution of accuracy on the test. What is most striking is the difficulty of the test. Not only is mean accuracy only one standard deviation above chance (50%) but no participant scored better than 75.0% correct. This high level of difficulty is desirable for a screening test of super-recognizers because very few participants are likely to achieve ceiling levels of accuracy. Also, given that subsequent performance on confirmatory tests is subject to the statistical phenomena of *regression to the mean*, potential super-recognizers with test scores that far exceed 2 SDs above the mean are more likely to achieve scores that exceed 2 SDs above the mean on subsequent confirmatory tests.

### Accuracy of all participant groups

The accuracy of the normative, lab and online samples are shown in Table 1 and Fig 3. Bonferroni comparisons (α = 0.0125) between each sample to the Normative data show no differences between the Normative sample and the Lab samples, but significantly higher accuracy in

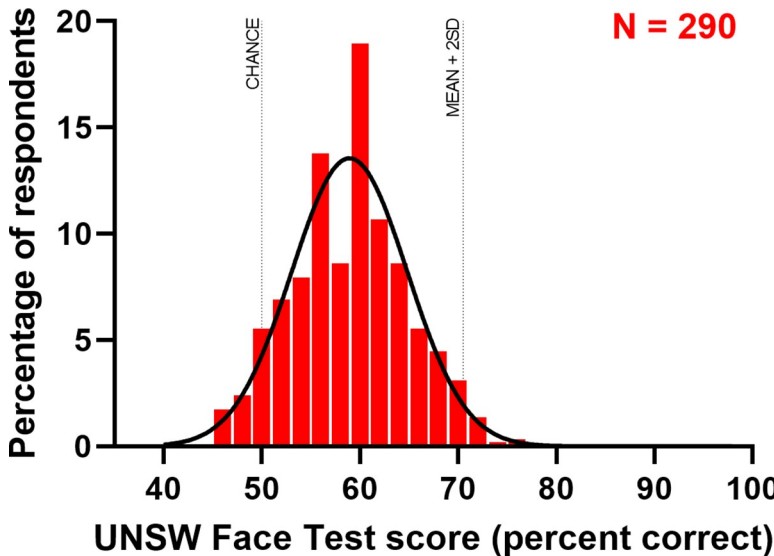

**Fig 2. Normative distribution of accuracy on the UNSW face test.**

the Online samples compared to the Normative sample, Lab Sample 1: Difference to normative = 1.3%, $p$ = .403, CI$_{95}$ [-0.69%, 3.34%]; Lab Sample 2: Difference to normative = 1.49%, $p$ = .172, CI$_{95}$ [-0.35%, 3.32%]; Online Sample 1: Difference to normative = 3.12%, $p < .001$, CI$_{95}$ [2.17%, 4.06%]; Online Sample 2: Difference to normative = 5.69%, $p < .001$, CI$_{95}$ [4.61%, 6.78%].

## Online sample accuracy

Closer inspection of performance distributions for the online samples is shown in Fig 4. These clearly show a shift in the distribution of scores for the online samples relative to the normative group (black distribution). As a result, we find a higher prevalence of participants meeting super-recognizer criteria in these samples, with 9% and 16% meeting super-recognizer criteria of 2 SDs above the mean for Online Samples 1 and 2, respectively. This shows that the recruitment strategies used to recruit participants for these tests–typically via popular media headlines such as '*So, you think you're good at recognizing faces*' (e.g. [46])–were effective in targeting high performers and is consistent with research showing that people have moderate insight into their face identification abilities [23, 47, 48].

Another important observation from Fig 4 is that despite over 20,000 attempts at the UNSW Face Test, no participant has achieved a perfect score (maximum accuracy = 93.3%).

**Table 1. Mean accuracy, standard deviations, and range of accuracy in each participant group.**

|  | n | Mean (%) | SD (%) | Range (%) |
|---|---|---|---|---|
| Normative Sample | 290 | 58.9 | 5.8 | 45.0–75.0 |
| Online Sample 1 | 22,776 | 62.0 | 6.4 | 40.8–93.3 |
| Online Sample 2 | 836 | 64.6 | 6.6 | 40.8–90.8 |
| Lab Sample 1[†] | 80 | 60.2 | 5.7 | 45.0–71.7 |
| Lab Sample 2 | 102 | 60.4 | 5.9 | 43.3–79.2 |

[†] Time 1 accuracy.

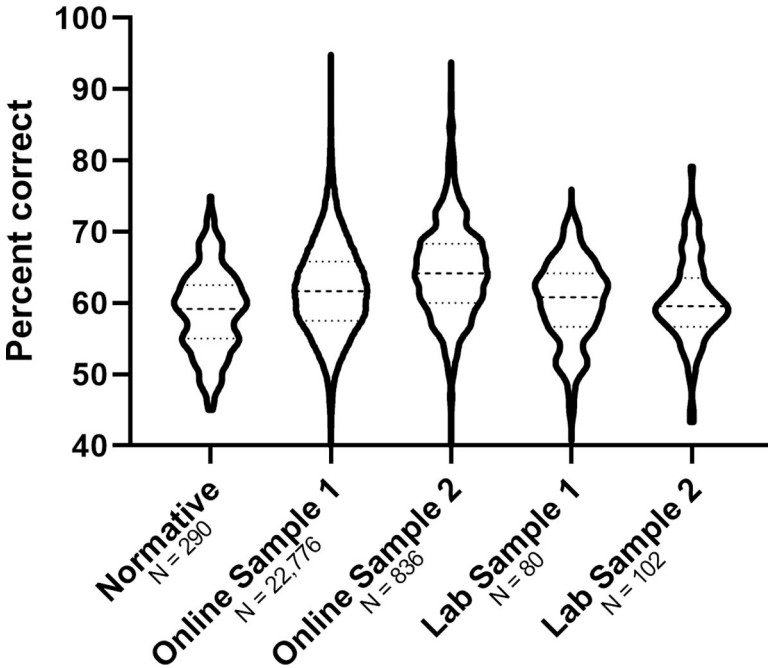

**Fig 3. Comparison of the normative, lab and online samples on UNSW face test scores.** The central dotted line indicates the mean and the lower and upper dotted lines indicate 25% and 75% percentiles, respectively.

This confirms the test does not suffer from ceiling effects and indicates that it is sufficiently difficult for the effective upper limit of human accuracy to be below the upper bound of its measurement range. The long tail of the distribution shows that the UNSW Face Test is sensitive to differences in performance up to 6 SDs above the mean.

## The effectiveness of the UNSW Face Test as a screening tool

The UNSW Face Test is designed to be an online screening tool for super-recognizers, so we examined how effective it is at finding the best performers on subsequent tests. The difficulty of the UNSW Face Test affords the opportunity to apply stricter criteria than the 2 SD cut-off that is typically used to classify super-recognizers. For example, of the 836 participants in Online Sample 2, 35 scored better than 3 SDs above the mean, a score that is well beyond the upper bound of both the CFMT+ and the GFMT.

Does this greater resolution at the top-end of the distribution provide benefits for a screening test? To answer this question, we analysed data from Online Sample 2, as this sample has 836 participants who completed the UNSW Face Test, CFMT+ and the GFMT (see Method for full details). We examined the accuracy of participants on the CFMT+ and GFMT, selected using progressively stricter selection criteria on the UNSW Face Test (1.7, 2, 2.5, and 3 SDs above the mean). We repeated this analysis using the CFMT+ and GFMT as screening tests. Results of this analysis are shown in Fig 5.

The top row of Fig 5 shows that the distribution of scores on the CFMT+ and GFMT improves as stricter criteria are applied on the UNSW Face Test. This same pattern is not evident when using the CFMT+ or GFMT for screening. For the CFMT, applying progressively stricter criteria does not select groups that perform progressively better on the other tests (Fig 5, middle row). For the GFMT, applying the strictest limit possible provides only moderate benefits (Fig 5, bottom row). These results show that the ability to set stricter screening criteria

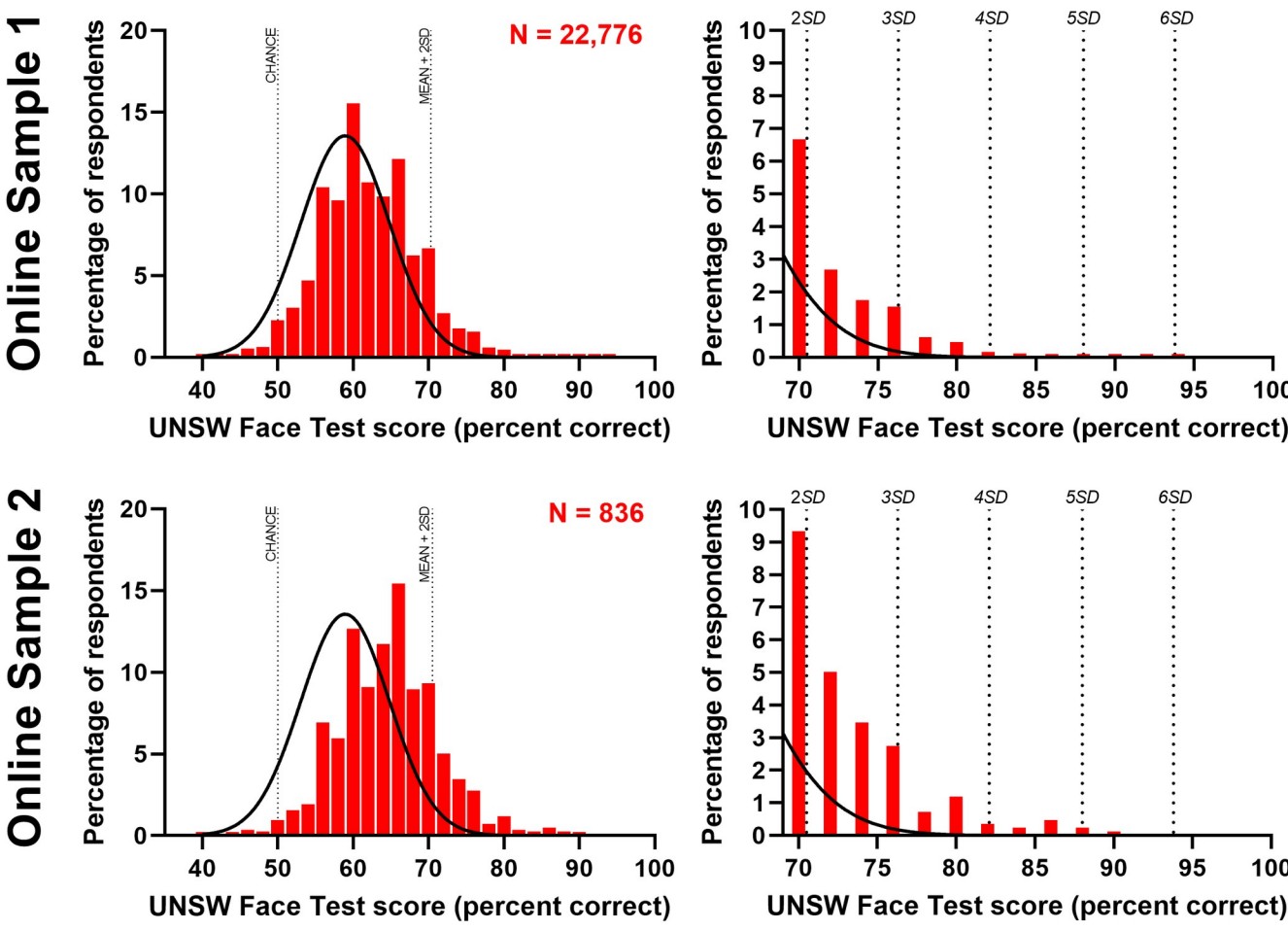

**Fig 4. Distribution of accuracy in online samples.** Left: Accuracy distribution of Online Sample 1 (top) and Online Sample 2 (bottom) compared to the normative accuracy distribution (black line). Right: Sample of distribution above the super-recognition threshold (2 SDs above the mean). The long tail of the distribution shows that the UNSW Face Test is sensitive to differences in performance up to 6 SD above the mean.

using the UNSW Face Test, compared to existing tests, provides researchers with an enhanced ability to target high performing people for follow-up testing.

Fig 6 shows the correlations between the three tests used to perform the analysis shown in Fig 5. Visual inspection of these figures suggests that the enhanced ability of the UNSW Face Test to screen for super-recognizers is due to the reduced frequency of ceiling level performance relative to the other tests. Ceiling effects in these tests are likely caused by the recruitment methods that explicitly targeted higher performers, which is consistent with the superior accuracy we observe in our online samples relative to lab-based samples.

## Test-retest reliability

To establish test-retest reliability of the UNSW Face Test, we used a lab sample because this gave us greater control over when participants completed the two testing sessions. Eighty participants in Lab Sample 1 completed the UNSW Face Test twice, one week apart (see Method for full details). Their scores at each time point are plotted in Fig 7. Test-retest reliability is $r(78) = 0.592$, $p < .001$, $CI_{95}$ [0.428, 0.718]), and relatively high in the context of psychometric tests more broadly, but slightly lower than the test-retest reliabilities of some existing face

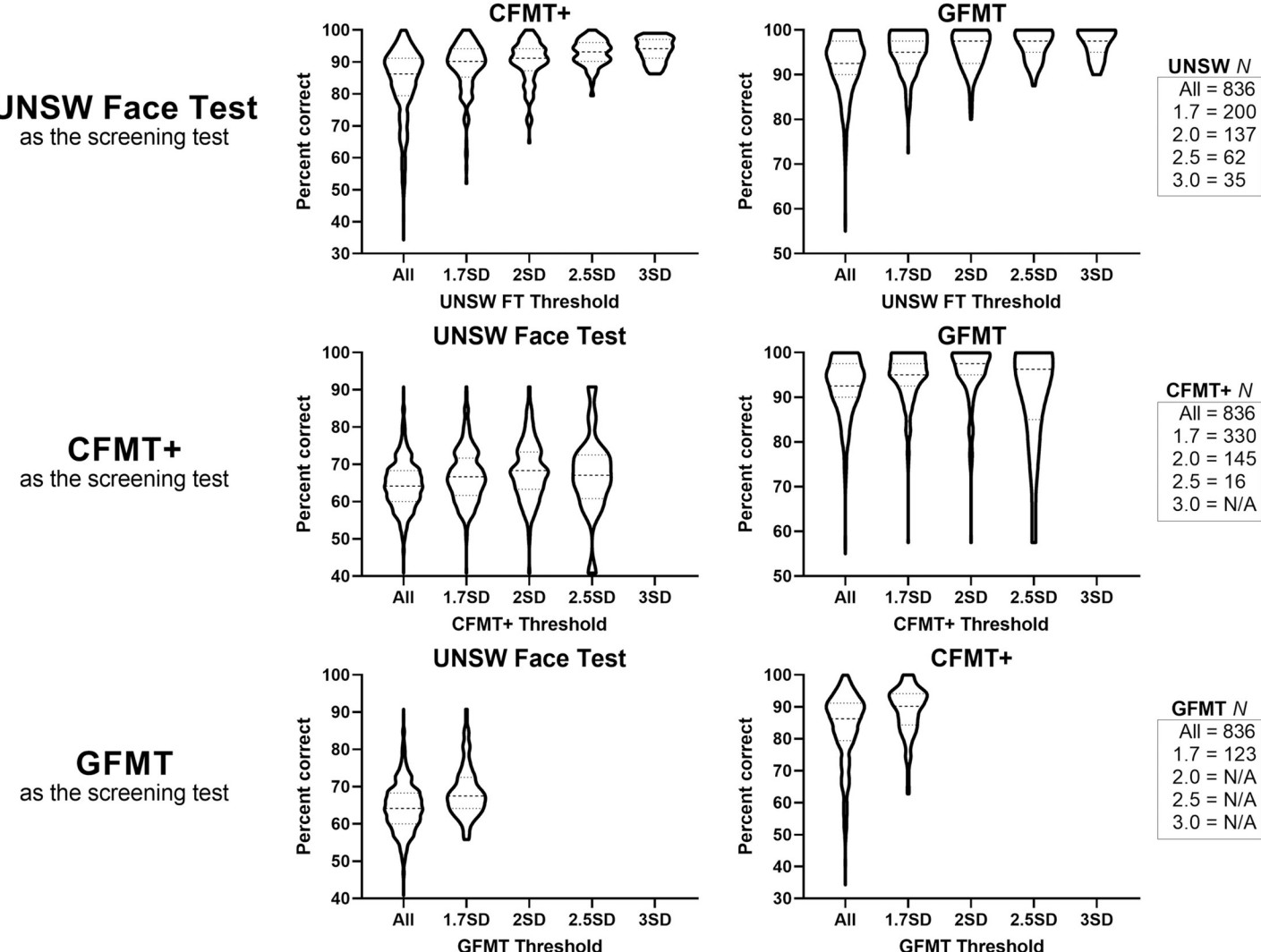

**Fig 5. Violin plots show how the distribution of performance on each of the online tests varies as a function of the screening criteria.** Each row shows the test used to select individuals (top row: UNSW Face Test; middle row: CFMT+; bottom row: GFMT). Boxes on the right show the number of participants in Online Sample 2 represented in each distribution. These data show that the ability to set stricter screening criteria on the UNSW Face Test provides greater precision for targeting high performing individuals to follow up testing than the CFMT+ or GFMT.

identification tests, including the CFMT (r = 0.70; [6]) and Kent Face Matching Test (r = 0.67; [49]). This might be attributable to the fact that accuracy on the UNSW Face Test is not calibrated to the midpoint of the measurement scale. Because our aim was to produce a challenging test with a greater resolution at the upper tail of the distribution, we were not concerned about compression of variance towards the lower end of the scale, but this is likely to limit the strength of test-retest correlations. Nonetheless, as demonstrated in Fig 5, the test is very effective at identifying high performers on subsequent tests.

We also note that repeating the UNSW Face Test significantly improved accuracy from 60.2% (SD = 5.7%) at Time 1 to 62.1% (SD = 5.8%) at Time 2, $t(79) = 3.30$, $p < .001$, $CI_{95} = [0.76\%, 3.07\%]$. This 1.9% improvement equates to 0.33 SDs, which means practicing this test buys only modest improvements that are unlikely to invalidate estimates of face identification ability.

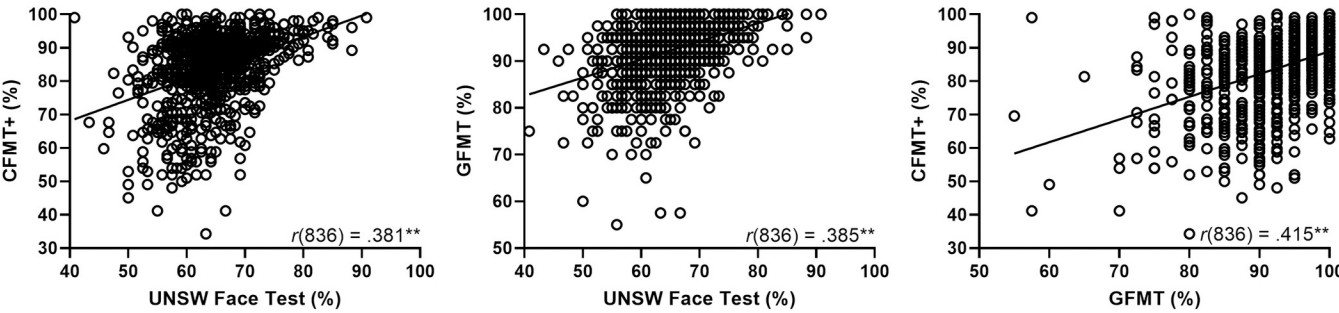

**Fig 6. Scatterplots of the correlations between the UNSW face test, CFMT+ and GFMT for Online Sample 2.** These results show the considerable variability in individual performance across the three tests, demonstrating the importance of repeated testing when establishing super-recognition. They also show that the UNSW Face Test does not suffer from ceiling effects, unlike existing tests and which can aid in the identification of super-recognisers. ** Significant at .01 level.

## Convergent validity

Next, we sought to establish convergent validity using a lab sample. Because online samples showed evidence of ceiling effects on the GFMT and CFMT+, we considered them unsuitable for this purpose (see Fig 6). 80 participants in Lab Sample 1 also completed the CFMT and GFMT at Time 1 (see Method for full details). Accuracy scores and correlations between the tests are shown in Table 2. Accuracy on the UNSW Face Test is more strongly correlated with the CFMT, indicating that the UNSW Face Test is more strongly aligned to face memory tasks like the CFMT compared to perceptual face matching tasks like the GFMT. This pattern of results becomes clear when comparing the associations between these tests with each UNSW Face Test sub-task (see also S2 Appendix). When analyzed by sub-task, the Memory task was significantly correlated with the CFMT+, $r(78) = 0.481$, $p < .001$, CI$_{95}$ [0.212, 0.634]), but not the GFMT, $r(77) = 0.191$, $p = .091$, CI$_{95}$ [-0.031, 0.346]). Conversely, the Sort task was correlated equally with the CFMT+, $r(78) = 0.399$, $p < .001$, CI$_{95}$ [0.197, 0.569]), and the GFMT, $r(77) = 0.336$, $p = .002$, CI$_{95}$ [0.124, 0.519]). This pattern of correlation supports those shown by

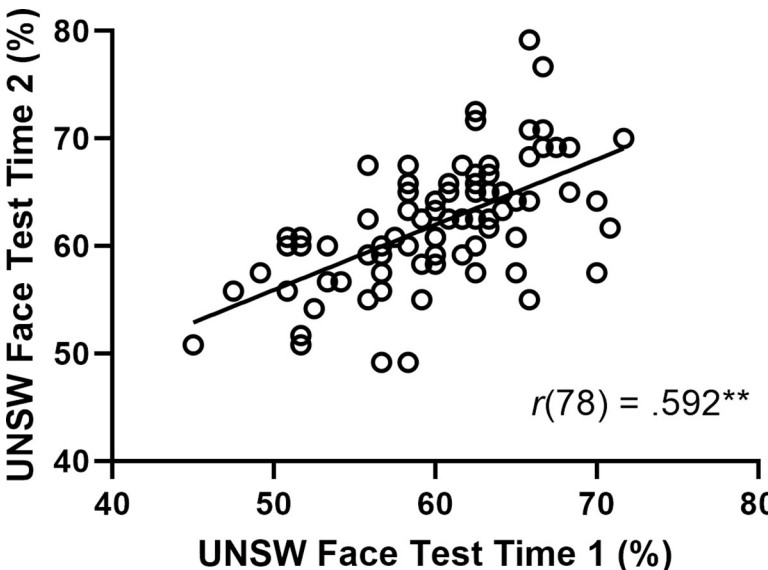

**Fig 7. Test-retest reliability on the UNSW face test after a one-week delay.** ** Significant at .01 level.

**Table 2. Reliable correlations between the UNSW Face Test and the CFMT and GFMT for Lab Sample 1 demonstrate high convergent validity.**

| | Mean (%) | SD (%) | UNSW FT Time 1 | | UNSW FT Time 2 | |
|---|---|---|---|---|---|---|
| | | | $r$ | CI$_{95}$ | $r$ | CI$_{95}$ |
| UNSW FT Time 1 | 60.2 | 5.7 | | | .592** | .428 - .718 |
| UNSW FT Time 2 | 62.1 | 5.8 | .592** | .428 - .718 | | |
| CFMT | 79.7 | 11.6 | .544** | .368 - .682 | .609** | .449 - .731 |
| GFMT | 71.9 | 12.3 | .359** | .150 - .537 | .259* | .040 - .454 |

* Significant at .05 level.

** Significant at .01 level.

overall accuracy, in that the UNSW Face Test is more strongly associated with face memory than matching ability. These correlations are consistent with previous reports of an association between standardized tests of face identification ability [11, 25, 27, 30, 49, 50] and provide evidence of high convergent validity.

## Discriminant validity

Discriminant validity of the UNSW Face Test was established using Lab Sample 2. This cohort consisted of 102 participants who completed the UNSW Face Test, the CFMT+, in addition to two non-face tasks: the Cambridge Car Memory Test, and the Matching Familiar Figures Test (see Method for full details). Their accuracy scores are shown in Table 3, along with correlation coefficients between each of the tests. Consistent with the results of the previous analysis, we find evidence of convergent validity: the UNSW Face Test is significantly positively correlated with the CFMT+, $r(100) = .306$, $p = .002$, CI$_{95}$ [0.119, 0.472]. More important for this analysis, we find evidence of discriminant validity: the UNSW Face Test does not correlate significantly with performance on either the Cambridge Car Memory Test, $r(100) = .034$, $p = .738$, CI$_{95}$ [-0.162, 0.227], or the Matching Familiar Figures Test, $r(100) = .142$, $p = .154$, CI$_{95}$ [-0.054, 0.328]. This pattern confirms that the UNSW Face Test has discriminant validity and is measuring domain-specific face identification abilities.

## Effects of participant demographics

To investigate demographic effects, we examined the effects of age, ethnicity, and gender on accuracy by combining Online Samples 1 and 2. We used our Online Samples for this analysis because large online cohorts are typically more diverse and heterogeneous than university student samples. The effects of age on accuracy were striking and described in detail below, whereas effects of ethnicity and gender were more subtle and so these are reported in S1 Appendix. Those using the UNSW Face Test may wish to calculate norms for specific

**Table 3. Mean accuracy and correlation matrix for all tests in Lab Sample 2, demonstrating discriminant validity.**

| | M (%) | SD (%) | Face identification tasks | | Non-face tasks | |
|---|---|---|---|---|---|---|
| | | | UNSW FT | CFMT+ | CCMT | MFFT |
| UNSW FT | 60.4 | 5.9 | | .306** | .034 | .142 |
| CFMT+ | 66.9 | 10.4 | .306** | | .191 | .265** |
| CCMT | 65.6 | 12.5 | .034 | .191 | | .329** |
| MFFT | 71.5 | 17.5 | .142 | .265** | .329** | |

** Significant at .01 level.

**Table 4. Mean accuracy (%) and standard deviation by participant age from online samples.**

| | Age group (years) | | | | | | | | | | | | | | | | | |
|---|---|---|---|---|---|---|---|---|---|---|---|---|---|---|---|---|---|---|
| | Adolescent | | | Young adult | | | Early middle age | | | 50–59 | | | 60–69 | | | 70–76 | | |
| | (10–17) | | | (18–35) | | | (36–49) | | | | | | | | | | | |
| | N | M | SD | N | M | SD | N | M | SD | N | M | SD | N | M | SD | N | M | SD |
| Overall | 1470 | 59.1 | 5.8 | 10163 | 62.8 | 6.5 | 6905 | 62.5 | 6.4 | 3149 | 61.1 | 5.9 | 1139 | 59.8 | 5.8 | 169 | 58.8 | 5.1 |
| Memory | 1470 | 62.2 | 8.4 | 10163 | 65.6 | 8.7 | 6905 | 65.3 | 8.7 | 3149 | 62.6 | 8.3 | 1139 | 60.5 | 8.0 | 169 | 59.5 | 7.3 |
| Sorting | 1470 | 57.5 | 6.8 | 10163 | 61.4 | 7.6 | 6905 | 61.1 | 7.7 | 3149 | 60.4 | 7.3 | 1139 | 59.5 | 7.2 | 169 | 58.4 | 6.5 |

demographics. All data is available in S2 Datasheet for this purpose, but to facilitate this process in Table 4 we provide the age-specific data from our online samples (see [51]).

Fig 8 shows accuracy on the UNSW Face Test as a function of age. Previous research shows that face identification ability increases markedly from childhood to adulthood–peaking at around age 31 –before slowly declining with further aging ([52]; see also [53]). Visual inspection of test accuracy in Fig 8 shows a strikingly similar pattern. We computed estimated peak accuracy and standard error by fitting a quadratic function to the logarithm of age and then using a bootstrapping resampling procedure, resampling from the data with replacement 200 times. The estimated age of peak accuracy for the UNSW Face Test is 30.7 years (SE = 0.2), which is remarkably close to the 31.4 years (SE = 0.5) for the CFMT reported by Germine and colleagues [52]. In both the CFMT and UNSW Face Test, the ages of the faces were young adults and so these results may reflect an own age bias whereby younger participants benefited from the choice of stimuli [54, 55]. However, because the average age of faces in the UNSW Face Test is almost 10 years younger than the peak performance age, this explanation alone cannot account for this age effect.

Visual inspection of Fig 8 suggests that accuracy on the memory sub-task is modulated more by participant age in comparison to the sorting sub-task. This is apparent both in terms of a greater increase in accuracy from childhood to adulthood (Memory increase = 9.3%,

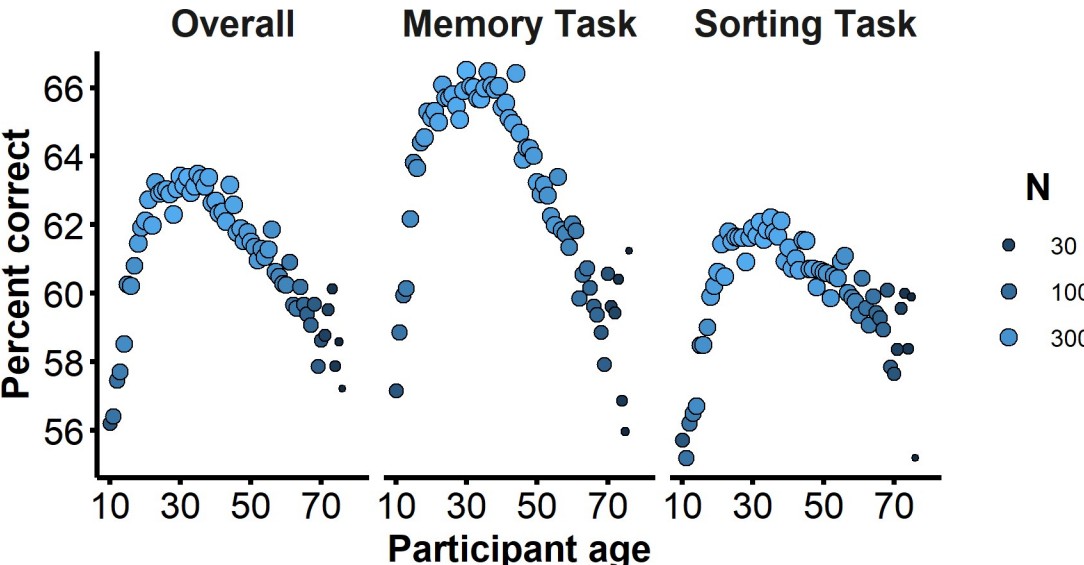

**Fig 8. Average accuracy for each participant age on the UNSW Face Test separated for overall (left), memory task (middle) and sorting task (right).** Size and shade of each data point show the number of participants in that age group.

SE = 0.4; Sorting increase = 7.7%, SE = 0.3), an earlier peak (Memory = 28.6 years, SE = 0.2; Sorting = 32.5 years, SE = 0.4), and a more marked decline of accuracy after the peak (Memory decrease = 8.0%, SE = 0.3; Sorting decrease = 4.0%, SE = 0.2). The divergence of aging effects for memory and sorting tasks is consistent with previous work showing that face identification accuracy is less sensitive to aging in the perceptual matching of face images, compared to recognition memory tasks [27].

## General discussion

The UNSW Face Test is a challenging new online screening test designed to identify super-recognizers. Unlike existing face identification tests, the UNSW Face Test is designed to be administered online and delivered *en masse* to large cohorts of participants. Because super-recognizers are rare, a mass screening tool enables researchers to identify larger groups of super-recognizers for follow-up confirmatory testing than is possible with existing tests. This will improve researchers' ability to achieve the statistical power necessary to investigate the cognitive, perceptual, and neural mechanisms supporting the highest levels of accuracy in face identification tasks.

An important property of the UNSW Face Test is that it is extremely challenging. Despite testing over 24,000 participants, no participant has achieved a perfect score at the time of writing. It is, therefore, an open question as to whether the limits of human ability in face identification tasks fall below the upper bound of the measurement scale used in this test. Moreover, because the accuracy of super-recognizers is below this upper-bound, it enables researchers to discriminate between super-recognizers that achieve, for example, a score that is 2 SDs vs 4 SDs. 4SD above the mean. Although the accuracy threshold used to define super-recognizers varies across studies, we have shown here that stricter recruitment criteria translate to higher performance in participant groups. So from a pragmatic viewpoint, if the goal is to study the highest performing participants on face identification tasks, adopting stricter criteria on screening tests will yield a greater proportion of high performing participants in follow up confirmatory tests.

The UNSW Face Test will enable researchers to adopt a more staged approach to recruiting super-recognizers. A great deal of effort is required to create a standardized psychometric test that is well-calibrated, reliable and provides a valid measure of ability [30]. And yet, many standardized tests are available online as initial screening tools, which means participants could practice the tests repeatedly. This is problematic because it reduces the legitimacy of these tests in scientific use and also, perhaps more concerningly, where these tests have been used in recruitment for security and policing roles [13, 18, 28].

We propose a solution whereby an initial screening test is followed by a battery of other standardized tests, held under stricter control, to verify an individual's ability. We have used the UNSW Face Test for precisely this purpose and have found it to be a very effective recruitment tool. Because the people pictured in the test have agreed for their images to be used, it can be linked to popular media content, and the interactive nature of the task is suited to engaging consumers of popular media. Anecdotal accounts from super-recognizers and our students suggest that they find the task enjoyable and are motivated to perform well. We attribute this to the difficulty of the test, and its strong *face validity*–stemming from the fact it was created using the type of images that people typically encounter on the internet.

Scientific understanding of superior ability in face identification is limited, and studies typically recruit small samples of super-recognisers ($n < 10$). Given that super-recognizers are increasingly being deployed to perform important real-world tasks, there are strong theoretical and practical motivations for researchers to rectify this in the years ahead [17]. We hope the

UNSW Face Test can support the initial recruitment phase on which these research activities are based, enabling researchers to find more individuals with this intriguing ability.

## Method

### Ethics statement

This research was approved by the Deputy Vice-Chancellor (Research) of the University of New South Wales on the advice of the Human Research Ethics Advisory Panel for data collected in Australia and by the University of Greenwich Research Ethics Committee for data collected in the United Kingdom. All participants gave their informed consent either digitally or in writing. The participation of individuals under age 18 on the UNSW Face Test without the express permission of a parent/guardian has been deemed to be consistent with the maturity and understanding requirements of the Australian government's National Statement on Ethical Conduct in Human Research. The faces of the individuals depicted in Fig 1 and those included in the test have given written informed consent (as outlined in PLOS consent form) to publish and use their faces for these purposes.

### Descriptions of tests

**Glasgow Face Matching Test (GFMT).** In the GFMT [27] participants decide whether pairs of images show the same person or two different people. Participants completed the short version of the GFMT, which contains 40 image pairs (20 match, 20 non-match). GFMT images were captured minutes apart in studio-conditions with different cameras.

**Cambridge Face Memory Test (CFMT).** The CFMT [26] is a standardized test of face memory. In this test, there are 3 blocks of 24 increasingly difficult trials. Participants memorize 6 novel faces (study phase) and then attempt to identify them in a three-person lineup (test phase). Images in block 1 are the same as those shown in the study phase. Images in block 2 are novel images of the study faces captured in untrained views and lighting conditions. Finally, images in block 3 are novel images that have been degraded with visual noise.

**Cambridge Face Memory Test–long form (CFMT+).** The CFMT+ [9] contains all trials from the original CFMT but also includes an additional, more difficult block of 24 trials intended to prevent ceiling effects. In this block, participants learn and are tested on novel images that contain more extensive visual noise and variability in the pose, expression, and visible features.

**Cambridge Car Memory Test (CCMT).** The CCMT [44] was created as a measure of individual differences in object discrimination. Using the same trial structure as the CFMT, this test provides a measure of object recognition ability that is independent of face recognition.

**Matching Familiar Figures Test (MFFT).** The MFFT [45] is a task that measures cognitive style, impulsivity versus reflexivity. Participants decide whether a target drawing is identical to one of six variants, or absent. Participants complete 20 trials.

### Participant samples

For all test samples, participants were excluded from the analysis if they failed any one of following criteria: 1) scored less than 40% overall on the test, 2) responded exclusively match or nonmatch in the Memory task, 3) did not move any images on a trial in the Sort task. With regards to participants' ethnicity, in all cases, ethnicity was self-identified by participants through a free-text response or by selecting from a drop-down menu with a selection of ethnic backgrounds that are common in Australia.

**Normative sample.** We recruited an online sample of 321 US residents from Amazon's Mechanical Turk who completed the UNSW Face Test. Thirty-one participants met the exclusion criteria and were removed from analyses. The final sample of 290 participants (114 males and 176 females, mean age = 37.1 years, SD = 11.4 years) contained 212 individuals of European descent (73.1%), 25 individuals of African descent (8.6%), 24 individuals of Asian descent (8.3%), 15 individuals of Hispanic descent (5.2%), 5 individuals of mixed descent (1.7%), 1 individual of Middle Eastern descent (0.3%), and 8 other/not-specified (2.8%).

**Online sample 1.** Participants in Online Sample 1 were 24,159 people who completed only the UNSW Face Test via www.unswfacetest.com between September 7, 2017, and August 23, 2019. Because this is an online test our sample likely contains multiple attempts by some participants. When we were able to link multiple attempts to the same email address we only included their first attempt. There were 1,383 participants who met the exclusion criteria and were removed from analyses. This left a final sample of 22,776 participants (9,211 males, 13,277 females, and 288 other/not specified, mean age = 36.7 years, SD = 13.9 years). The sample consisted of 17,241 individuals of European descent (75.7%), 2157 individuals of Asian descent (9.5%), 1,330 individuals of mixed descent (5.8%), 892 other (3.9%), 405 individuals of Pacific Island descent (1.8%), 220 individuals of Middle Eastern descent (1%), 196 individuals of Hispanic descent (0.9%), 178 individuals of Aboriginal Australian descent (0.8%), and 157 individuals of African descent (0.7%).

**Online sample 2.** Participants in Online Sample 2 were 866 people who completed the CFMT+, GFMT and UNSW Face Test online, in this fixed order. These participants volunteered to complete the tests online by clicking an advertisement for research participation with The University of Greenwich between March 7, 2018, and August 1, 2019. There were 30 participants met the exclusion criteria and were removed from analyses. The final sample consisted of 836 participants (355 males, 456 females, and 25 other/not specified, mean age = 34.6 years, SD = 11.5 years) comprising 682 individuals of European descent (81.6%), 60 individuals of Asian descent (7.2%), 54 individuals of mixed descent (6.5%), 15 other (1.8%), 14 individuals of Hispanic descent (1.7%), 9 individuals of African descent (1.1%), 1 individual of Pacific Island descent (0.1%) and 1 individual of Middle Eastern descent (0.1%).

**Lab sample 1.** Participants in Lab Sample 1 completed two test sessions, one week apart. At Time 1, participants completed the GFMT, CFMT and UNSW Face Test, in this fixed order. At Time 2, participants completed the UNSW Face Test again. Participants took between 45–60 minutes to complete the test battery at Time 1 and 15–20 minutes to complete Time 2. This sample consisted of 80 UNSW undergraduate students (21 males and 59 females, mean age = 19.3 years, SD = 3.0 years) who participated in exchange for course credit. No participants met the exclusion criteria. There were 46 individuals of Asian descent (57.5%), 23 individuals of European descent (28.8%), 6 individuals of mixed descent (7.5%), 3 others (3.8%), and 2 people of Middle Eastern descent (2.5%). Due to time constraints, one participant did not complete the GFMT but did complete the other tests, resulting in a final sample of 79 for the GFMT and 80 for the remaining tests.

**Lab sample 2.** Lab Sample 2 completed the following test battery in a counterbalanced order: the UNSW Face Test, CFMT+, CCMT, and the MFFT. Participants took between 75–90 minutes to complete the test battery. This sample consisted of 102 UNSW undergraduate students (27 males and 75 females, mean age = 18.9 years, SD = 1.6 years) who participated in exchange for course credit. No participants met the exclusion criteria. There were 62 individuals of Asian descent (60.8%), 31 individuals of European descent (30.4%), 3 individuals of mixed descent (2.9%), 2 individuals of African descent (2%), 2 individuals of Aboriginal Australian descent (2%), and 2 people of Middle Eastern descent (2%).

## Supporting information

**S1 Appendix. Additional demographics analysis.**
(DOCX)

**S2 Appendix. Analysis by subtask.**
(DOCX)

## Acknowledgments

Thanks to Daniel Noble and Natalija Pavleski for assistance with stimuli selection and preparation, to Christel Macdonald, Daniel Guilbert, Albert Lin, and Monika Durova for assistance with data collection, and to Richard Kemp for his thoughtful commentary.

## Author Contributions

**Conceptualization:** James D. Dunn.

**Data curation:** James D. Dunn.

**Formal analysis:** James D. Dunn.

**Funding acquisition:** David White.

**Investigation:** James D. Dunn, Stephanie Summersby, Alice Towler, Josh P. Davis, David White.

**Methodology:** James D. Dunn, Stephanie Summersby, Alice Towler, Josh P. Davis, David White.

**Project administration:** James D. Dunn, Stephanie Summersby, Alice Towler, David White.

**Resources:** James D. Dunn, Stephanie Summersby, David White.

**Software:** James D. Dunn.

**Supervision:** David White.

**Visualization:** James D. Dunn.

**Writing – original draft:** James D. Dunn.

**Writing – review & editing:** James D. Dunn, Stephanie Summersby, Alice Towler, Josh P. Davis, David White.

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
