## [Decision Letter · Decision Letter 0]

28 Jul 2020

PONE-D-20-16507

UNSW Face Test: A screening tool for super-recognizers

PLOS ONE

Dear Dr. Dunn,

Thank you for submitting your manuscript to PLOS ONE. After careful consideration, we feel that it has merit but does not fully meet PLOS ONE’s publication criteria as it currently stands. Therefore, we invite you to submit a revised version of the manuscript that addresses the points raised during the review process.

As you can see there are various wide ranging opinions from the Reviewers, however, broadly speaking there is a positive disposition to this paper. The most important areas to work on are to improve the precision in the language throughout the manuscript and ensure that every claim made is based on the data collected. I will not reiterate the Reviewer's comments as they are quite detailed.

We look forward to receiving your revised manuscript.

Kind regards,

Peter James Hills, PhD

Academic Editor

PLOS ONE

Journal Requirements:

2. Please provide additional details regarding participant consent, and whether this was obtained from participants for all parts of your study. In the Methods section, please ensure that you have specified (1) whether consent was informed and (2) what type you obtained (for instance, written or verbal). If your study included minors, state whether you obtained consent from parents or guardians. If the need for consent was waived by the ethics committee, please include this information.

3. We note that Figure [1] includes an image of a participant in the study]. 

4. Please note that according to our submission guidelines (http://journals.plos.org/plosone/s/submission-guidelines), outmoded terms and potentially stigmatizing labels should be changed to more current, acceptable terminology. For example: “Caucasian” should be changed to “white” or “of [Western] European descent” (as appropriate).

Reviewers' comments:

Reviewer's Responses to Questions

**Comments to the Author**

1. Is the manuscript technically sound, and do the data support the conclusions?

Reviewer #1: Yes

Reviewer #2: Yes

Reviewer #3: Yes

Reviewer #4: Yes

Reviewer #5: Yes

Reviewer #6: No

2. Has the statistical analysis been performed appropriately and rigorously? 

Reviewer #1: Yes

Reviewer #2: Yes

Reviewer #3: Yes

Reviewer #4: Yes

Reviewer #5: Yes

Reviewer #6: No

3. Have the authors made all data underlying the findings in their manuscript fully available?

Reviewer #1: Yes

Reviewer #2: Yes

Reviewer #3: Yes

Reviewer #4: Yes

Reviewer #5: Yes

Reviewer #6: No

4. Is the manuscript presented in an intelligible fashion and written in standard English?

Reviewer #1: Yes

Reviewer #2: Yes

Reviewer #3: Yes

Reviewer #4: Yes

Reviewer #5: Yes

Reviewer #6: Yes

5. Review Comments to the Author

Reviewer #1: This manuscript introduces a new test, the UNSW-Face Test that can be used for online screening of super recognizers – people with extraordinarily good face recognition ability. The test has good psychometric properties for this purpose, is correlated with other measures of face recognition, and not correlated with a car recognition test.

The UNSW-FT looks to be a useful addition to the existing set of standardized face tests. I have only minor comments which I have listed below.

The authors didn't report many separate analyses of the old-new vs sorting task data, but it seems there might be some additional interesting differences between the two sub-tests. If there are, maybe those analyses could be placed into an appendix or the supporting materials.

Line 27: “that” should be “than”.

L64: The authors mention that existing tests are unsuitable for online testing, but the CFMT at least has been used successfully in a number of online studies (Wilmer et al., 2010; Shakeshaft et al., 2015). Are there other reasons that make the new test more suitable than existing tests?

L241: Do the authors have RT data to assess whether individual participants' RTs were so long it suggests they were using an unusual strategy (e.g., taking photos of the faces)? RTs might provide a means to jettison questionable participants.

L313: The authors suggest the UNSW-FT is more strongly correlated with face memory measures and that is plausible, but differences in the sensitivity and reliability of the CFMT and the GFMT may also contribute to the stronger correlations with CFMT. Do the authors have evidence or does data exist (e.g., pre-existing data on reliability of CFMT vs GFMT) that speaks to the influence of these different factors on the correlations?

L440: Since the M-Turk participants were all from the US, it doesn't sound right to call them "Africans", "East Asian". I see the analogy to "Caucasians", but that is commonly used to refer to an ethnic group whereas "Africans" sounds like someone from Africa to my ears regardless of their ancestral history (e.g. a white person from Kenya).

Reviewer #2: The manuscript introduces the UNSW Face Test, an online test of face processing ability as an initial screening tool for SRs. The analyses presented aimed to show that the test 1) is of sufficient difficulty to minimise ceiling effects, 2) shows consistency with current tests used to identify SRs, 3) captures performance specific to face processing and not other general cognitive ability, and 4) has retest reliability.

The manuscript was written in a clear and concise manner and I feel that the aims have been achieved. I only have the following minor comments which the authors may find useful.

Minor comments:

1)Line 64: It is unclear why existing tests are unsuitable for online testing.

2)Line 88 onwards: I get the point about general ability, but it seems that the two components of the test were designed to capture different aspects of face processing. Might there be any utility/interest in looking at performance across the two components separately, given that different applications require an individual to be good at specific types of processing?

3)Line 127: What were the criria used to select the foils?

4)Line 133: I'm not 100% certain of the technical details of image processing, but I think the specification of image resolution is only useful if all original images were resampled from a higher resolution. Might be good to make it clear that the image quality for all images were generally similar.

5)Effects of participant demographic: Since the stimuli used were images of undergraduate students and most are likely to be within the Young adult -Early middle age. Maybe this could have influenced the age effect.

6)I assume the same exclusion criteria as the online sample were used for the other samples?

7)I might have missed this, but I can only see ethics details for the use of the stimuli. Of particular concern is that data from the adolescent age group is reported for the online samples. I'm not familiar with the data protection laws of the different countries, but wanted to highlight this in case it might be an issue.

Reviewer #3: The authors here provide a new test of face recognition ability (UNSW face test) that can be used to help detect super-recognizers). The test is freely available for researchers to use. Normative data and demographic norms are provided. The authors report significant correlations between the UNSW test and the CFMT + and GFMT, and argue that the UNSW face test can be used to detect general face recognition ability. A new, challenging, test of face recognition ability is an important development to progress the field of face recognition. I believe that the test will be a useful resource to advance the study and recruitment of super-recognizers.

Comments

1. Line 171, description of participant groups. The wording ‘targeting high performers on the GFMT and CFMT+’ suggested to me that all online participant groups had either already completed or would complete the CFMT and GFMT. After reading the methods section at the end of the manuscript my understanding is that this is not the case. Please refine the explanation here.

2. Line 193 – normative data. Was there a reason why normative data was based on the data from mechanical Turk instead of from other participant groups? Norm is lower than online samples. Potential issues here if lower norm used in SR recruitment if mechanical Turk not best matched normative group. Could be linked to motivation but arguably applied SRs would be more similar to the most motivated control group.

3. Line 308. Is there a reason why convergent reliability is based only on lab sample when online sample 2 also completed CFMT and GFMT?

4. In text report of Fig 6. Suggest that reference to variation in performance at individual level is added – these differences are apparent in the scatterplots, but only group level correlations addressed in text. Acknowledgement of variability in individual level results important especially for applied use of the screening test. Would also be relevant to acknowledge individual spread if this was also found for results reported in Table 2 (through in text comment or addition of scatterplot).

5. Line 312 – UNSW face test more strongly correlated with CFMT. It would be interesting to see the breakdown of correlations for Part 1 and Part 2 of the UNSW face test with GFMT and CFMT+. Did part 1 correlate more strongly with CFMT as these tests are more similar?

6. Line 338 – explained why the sample was used. I found this sentence really helpful and would suggest that where sensible similar sentences are added for the above results sections as it was sometimes confusing to keep track of which group was used and why others were not included.

Minor

- Line 27, typo. ‘that’ -> ‘than’

- Line 385. Helpful to add context for why criteria can vary (different operational context etc). Otherwise reader may be unsure about the level of performance that is required to be defined as an SR. A sentence on the lack of clarity over definition of SR may suffice.

Reviewer #4: This is a good paper that makes a valuable contribution to the field. I expect that the UNSW Face Test will be widely adopted and that the paper will attract many citations. I recommend publication. Two major strengths of the new test, given its purpose, are the unusually high level of difficulty and the use of ambient images. The inclusion of test-retest reliability and relation to other cognitive tests make the scores much easier to interpret.

The abstract would be improved by the following minor changes:

- change "available for free for scientific use" to "available free for scientific use"

- change "an important tool for their initial recruitment" to "an important step in initial recruitment"

- change "before completing confirmatory testing" to “before confirmatory testing”

- change “normative data on the test” to “normative data on our screening test”

- change "stricter selection criteria that" to "stricter selection criteria than"

Other comments:

Page 3. Line 55. “identifying super-recognizers based on self-report alone is unreliable”. Zhou & Jenkins (2020) found that high performers consistently underestimate their performance, both in absolute terms (estimated accuracy) and in relation to others (estimated rank).

Zhou, X., & Jenkins, R. (2020). Dunning–Kruger effects in face perception. Cognition, 203, 104345.

Page 3. Line 56. Change “This coupled with” to “This constraint, coupled with”

Page 4. Lines 64–66. “However, existing standardised tests of face identification ability are unsuitable for online testing. For example, the CFMT and GFMT are carefully calibrated psychometric tests intended to be reliable measures of a person’s ability.” It would be helpful to briefly explain why these properties make the tests unsuitable for online testing. They sound like desirable properties on the face of it.

Page 4. Lines 69–75. This argument is not well developed. In particular, the sentence beginning “Further” seems to run against the preceding sentence. The paragraph would benefit from a thoughtful rewrite.

Page 7. Line 136. Change “Participants’ scores” to “A participant’s score”

Page 8. Lines 155–162. Avoid excessive repetition of “participants”. e.g. “Next, they sort”, “resulting in a maximum score of 80”.

Page 10: Lines 214–217. Briefly explain what the percentage values in the main text refer to, and their relation to Table 1.

Page 14. Lines 296–298. One possible explanation is that the high task difficulty forced participants to guess on a high proportion of trials. There is no reason that ‘pure’ guesses should be correlated across attempts.

Page 17. Lines 353–355. This is an interesting result. It could be taken to mean that face recognition ability peaks around age 30. However, it could also be taken to mean that it is easier to recognise faces that are (roughly) your own age (i.e. the ‘other-age effect’). I don't think the current data, on their own, can distinguish these two possibilities. It might be worth briefly commenting on the ambiguity, if only for the sake of future systematic reviewers and meta-analysts.

Reviewer #5: In this very interesting study the authors present an online test for screening large samples in order to identify persons with extremely high face identity processing skills (“super-recognizers”). Data were collected from a total of five samples, two of these were tested in the lab. Overall, the entire data set includes ~24,000 participants. Some samples additionally performed other face tests (Cambridge Face Memory Test; Glasgow Face Matching Test), as well as tests tapping into other, i.e. non face-related tasks (Cambridge Car Memory Test; Matching Familiar Figures Test). The authors conclude from the data that their “UNSW Face Test” is difficult, not prone to ceiling effects, largely specific to face processing and therefore well suited to pre-select potential super-recognizers online, which can then be further tested in the lab for confirming or ruling out face super-recognition skills. This tool would be useful both for basic research and applied purposes.

Overall, I think the authors present and offer (the test is freely available for other researchers and institutions) an extremely useful resource, from which many labs can benefit. The manuscript is clearly and concisely written, and the design and the results are straightforward. Nevertheless, I have some questions and comments, which are outlined below:

Major (note that page numbers refer to manuscript pages, not the complete pdf file):

Page 8, line 161: It would be nice if there were not only results for overall accuracy, but also for hits and false alarms.

Page 10, line 10: Please add infos on sample size to the legend of Fig. 2.

Page 10, line 214: It is pretty obvious, but it would still make reading easier if it was made clear that the percentages refer to the differences.

Page 14, line 291: Test-retest reliability is not that great, and might be distorted by using the same version within one week. Maybe the authors could avoid potential repetion effects by calculating test-retest reliabilty for (post-hoc) contructed paralell versions?

Page 15, table 2: Isn’t it surprising that the correlation between UNSW FT Time 2 and CFMT is (numerically) larger than the correlation between UNSW FT Time 1 and UNSW FT Time 2? At the same time, I was surprised that the correlation between UNSW FT and CFMT+, which is an established tool for identifying super-recognizers, is rather low, and lower than for the “normal” CFMT. Do the authors have any explanations for these findings?

Page 20, line 435: Why was it decided to take the normative sample from US residents, when all other samples were from Australia?

Page 20, line 440: I think the categorization of participants is not precise and in some cases probably wrong: Is a person, whose ancestors were born and have lived for generations out of Africa, automatically “African”? What is meant here is probably the looks. Along these lines, the term “ethnicity” may also be wrong in this context, because it includes a cultural component. Of course, in real life, there is often an association between physical facial appearance and cultural background, but there is no causal relationship. Using the terms race (which is per se a problematic and inprecise term) and ethnicity interchangably, when acutally talking about the physical appearance of a person, means equalizing culture and biology, which seems very problematic to me.

Page 21, line 452: I would also pass a similar criticism (i.e. lack of precision) on the term “mixed-race”? How much and what kind of mix has to be there to count as “mixed-race”? In most regions of the world most people are more or less mixed, isn’t it?

Minor:

Page 5, line 102: delete “and” (or add “were” after “and”)

Page 6, line 119: add group or sample after “demographic”

Page 6, line 119: “for” example (rather than “or”)

Reviewer #6: See review comments in the attached document.

6. PLOS authors have the option to publish the peer review history of their article (what does this mean?). If published, this will include your full peer review and any attached files.

Reviewer #1: No

Reviewer #2: **Yes: **Nabil Hasshim

Reviewer #3: No

Reviewer #4: No

Reviewer #5: No

Reviewer #6: **Yes: **Meike Ramon & Jeff Nador

---

## [Author Response · Author response to Decision Letter 0]

10 Sep 2020

James D. Dunn, Stephanie Summersby, Alice Towler, Josh P. Davis, and David White 

PLOS ONE

Author response to editor and reviewers manuscript PONE-D-20-16507

UNSW Face Test: A screening tool for super-recognizers

We have copied reviewers’ comments below with our point by point responses denoted by “*”. All page and line references refer to the tracked changes version.

Journal Requirements:

*We have made changes to the title page to match the journal style requirements.

2. Please provide additional details regarding participant consent, and whether this was obtained from participants for all parts of your study. In the Methods section, please ensure that you have specified (1) whether consent was informed and (2) what type you obtained (for instance, written or verbal). If your study included minors, state whether you obtained consent from parents or guardians. If the need for consent was waived by the ethics committee, please include this information.

*We have provided the following ethics statement on pg 24 ln 513:

This research was approved by the Deputy Vice-Chancellor (Research) of the University of New South Wales on the advice of the Human Research Ethics Advisory Panel. All participants gave their informed consent either digitally or in writing. The participation of individuals under age 18 on the UNSW Face Test without the express permission of a parent/guardian has been deemed to be consistent with the maturity and understanding requirements of the Australian government’s National Statement on Ethical Conduct in Human Research. The faces of the individuals depicted in Fig 1 and those included in the test have given written informed consent (as outlined in PLOS consent form) to publish and use their faces for these purposes.

3. We note that Figure [1] includes an image of a participant in the study]. 

*We have updated Figure 1 and can confirm we have signed consent to publish these images for all of the individuals included. We have also added the consent for publication to the ethics statement (pg 24 ln 513).

4. Please note that according to our submission guidelines (http://journals.plos.org/plosone/s/submission-guidelines), outmoded terms and potentially stigmatizing labels should be changed to more current, acceptable terminology. For example: “Caucasian” should be changed to “white” or “of [Western] European descent” (as appropriate).

* We have updated our ethnicity labels throughout the manuscript and supplementary materials as suggested (e.g. pg 8 ln 157 and pg 26 ln 560). We note that references to race and/or ethnicity are not judged by us but self-identified in all cases and have noted that in the manuscript on pg 8 ln 157 for the test stimuli and on pg 25 ln 554 for the participants.

*We have included captions for our supporting information files on pg 30 ln 661.

Review Comments to the Author

Reviewer #1 

The authors didn't report many separate analyses of the old-new vs sorting task data, but it seems there might be some additional interesting differences between the two sub-tests. If there are, maybe those analyses could be placed into an appendix or the supporting materials.

*As suggested, we have repeated the analyses from the main text but separated by sub-task in S2 Appendix and include this data in S1 Dataset and report it in the main text on pg 18 ln 396. In summary, the Memory task correlates more with the CFMT+ than the GFMT, implying greater reliance on face memory, while the Sort task correlates equally with both of these tests, implying equal reliance on both matching and memory.

Line 27: “that” should be “than”.

*Corrected

L64: The authors mention that existing tests are unsuitable for online testing, but the CFMT at least has been used successfully in a number of online studies (Wilmer et al., 2010; Shakeshaft et al., 2015). Are there other reasons that make the new test more suitable than existing tests?

*We were not sufficiently clear on this point so have clarified our argument in the manuscript on pg 4 ln 72. We did not mean to imply that tests like the CFMT should not be used online. We have made changes throughout the manuscript to explain that the purpose of the UNSW Face Test. As this test an online screening test, using it can help retain existing psychometric tests for use as confirmatory testing or to use in the type of relatively constrained online recruitment that was carried out in the studies by Wilmer or Shakeshaft and colleagues (as opposed to unconstrained recruitment via a test link embedded in a news article). 

L241: Do the authors have RT data to assess whether individual participants' RTs were so long it suggests they were using an unusual strategy (e.g., taking photos of the faces)? RTs might provide a means to jettison questionable participants.

* We have data on the duration each participant spent on a trial and this data could be used as exclusion criteria if researchers wish. In response to another reviewer’s suggestion, we have included summary RT data in S1 Dataset and analysis of RT in S2 Appendix. We have explored our own dataset for whether RT should be used to remove some participants. For the normative data, we are confident that high RT would not explain any individual’s performance. For lab based data, we also are confident strategies like these were not used because participants were monitored by an experimenter. For online data, given the sheer quantity of data, we have decided against the analysis of the RT data in this instance. We expect that the use of strategies like taking photos is likely to be in the minority and therefore is unlikely to affect the outcome of the analyses we performed with these samples. 

L313: The authors suggest the UNSW-FT is more strongly correlated with face memory measures and that is plausible, but differences in the sensitivity and reliability of the CFMT and the GFMT may also contribute to the stronger correlations with CFMT. Do the authors have evidence or does data exist (e.g., pre-existing data on reliability of CFMT vs GFMT) that speaks to the influence of these different factors on the correlations?

*For the CFMT published data shows test-retest reliability of 0.7 (Wilmer et al., 2010), split-half reliability of 0.91 and a Cronbach’s alpha of 0.90 (Wilmer et al., 2012). For the GFMT, we only know of unpublished data from our lab which shows test-retest reliability of r(86) = .591, p < .001 (Tyler et al., in prep.). The upper limit for associations between any two tests is the geometric mean of the reliabilities of two measures, and so the upper limit of correlation between UNSW-FT and the CFMT is slightly higher (.65) than for the GFMT (.59). This may explain some of the differences. We now include a more detailed breakdown of the correlation between GFMT, CFMT and the memory and sorting subtasks of the UNSW-FT (pg 18 ln 396 - see also response to Reviewers 4 and 5). This analysis shows that the higher correlation between UNSW-FT and CFMT is specific to the memory subtask of the UNSW-FT, which supports our suggestion that the CFMT is correlated more with face memory measures. 

L440: Since the M-Turk participants were all from the US, it doesn't sound right to call them "Africans", "East Asian". I see the analogy to "Caucasians", but that is commonly used to refer to an ethnic group whereas "Africans" sounds like someone from Africa to my ears regardless of their ancestral history (e.g. a white person from Kenya).

*This point was also raised by the editor and we agree. We have ameded our descriptions of participant and face ethnicity throughout the manuscript. We also note that references to race and/or ethnicity are not judged by us but self-identified in all cases and have noted that in the manuscript on pg 8 ln 157 for the test stimuli and on pg 25 ln 554 for the participants.

Reviewer #2 

Minor comments:

1)Line 64: It is unclear why existing tests are unsuitable for online testing.

*We have clarified this point in the manuscript on pg 4 ln 72. Please see our response to Reviewer 1 for more details.

2)Line 88 onwards: I get the point about general ability, but it seems that the two components of the test were designed to capture different aspects of face processing. Might there be any utility/interest in looking at performance across the two components separately, given that different applications require an individual to be good at specific types of processing?

*This point was also raised by Reviewer 1. In response, we report additional analyses of the two sub-tasks in the manuscript on pg 18 ln 396 and in S2 Appendix. We also include this data separated by sub-task in S1 Dataset. In summary, the Memory task correlates more with the CFMT+ than the GFMT, implying greater reliance on face memory, while the Sort task correlates equally with both of these tests, implying equal reliance on both matching and memory.

3)Line 127: What were the criria used to select the foils?

*We have added the following description of foil selection to pg 6 ln 152 to clarify this step.

“Images in the UNSW Face Test were selected from a database of 236 consenting undergraduate students at UNSW Sydney (for full details see [38]). From this database, we selected 40 ‘target’ identities and a similar-looking foil identity. The mean age of the faces was 19 years old (SD = 1.4) and there was an equal number of males and females. Target-foil pairs were selected collaboratively by the research team whose goal was to find the individuals that were the most similar in facial appearance to each other in the database. Because we were designing a test to identify super-recognizers in Australia, we chose faces to be heterogeneous with respect to ethnicity and that reflect the diversity of the Australian population. Half the individuals self-identified as being of European descent. The other half self-identified as being as either Asian descent (5 were used as targets in the Memory task, 10 in the Sort task) or of Indian descent (5 as targets the Memory task). We then selected 5 images of each target and foil face.” 

4)Line 133: I'm not 100% certain of the technical details of image processing, but I think the specification of image resolution is only useful if all original images were resampled from a higher resolution. Might be good to make it clear that the image quality for all images were generally similar.

*We have added clarification in pg 8 ln 189 regarding the native resolution of images in the test. Many of the images were resampled from lower resolution, given the quality of the ambient images varied considerably, but we set a lower linit on resolution of original images as 60 pixels between the eyes. We believe this is a strength of the test as it represents the challenging image matching decisions that are faced in many applied settings (e.g. when comparing CCTV images). 

5)Effects of participant demographic: Since the stimuli used were images of undergraduate students and most are likely to be within the Young adult -Early middle age. Maybe this could have influenced the age effect.

*We have updated the manuscript on pg 21 ln 470 to include this argument. In both the CFMT and UNSW Face Test, the ages of the faces were young adults and so it is possible these results reflect an own age bias whereby younger participants were benefited by the choice of stimuli. However, because the average age of faces in the UNSW Face Test (19 years old) is almost 10 years younger than the peak performance age, this explanation alone cannot account for the observed age effect.

6)I assume the same exclusion criteria as the online sample were used for the other samples?

*Yes the same exclusion criteria were applied to all samples. This criteria meant that only participants in the online samples were removed. We have clarified the exclusion criteria in the participants' section (pg 25 ln 551).

7)I might have missed this, but I can only see ethics details for the use of the stimuli. Of particular concern is that data from the adolescent age group is reported for the online samples. I'm not familiar with the data protection laws of the different countries, but wanted to highlight this in case it might be an issue.

*We appreciate the author pointing this out. As directed by the editor, we have now provided an ethics statement on pg 24 ln 513. With regards to the data of the younger participant groups, the participation of individuals under age 18 in the UNSW Face Test without express permission of a parent/guardian has been deemed to be consistent with the maturity and understanding requirements of the Australian government’s National Statement on Ethical Conduct in Human Research where this research took place.

Reviewer #3 

Comments

1. Line 171, description of participant groups. The wording ‘targeting high performers on the GFMT and CFMT+’ suggested to me that all online participant groups had either already completed or would complete the CFMT and GFMT. After reading the methods section at the end of the manuscript my understanding is that this is not the case. Please refine the explanation here.

*We agree this was an inaccurate description of our approach for the online samples and have removed this line from the manuscript (pg 10 ln 240). It was only in online sample 2 that participants completed the UNSW Face Test, CFMT+ and GFMT and we examined using each of the tests to screen for super-recognisers in the other (i.e. “The effectiveness of the UNSW Face Test as a screening tool” analysis).

2. Line 193 – normative data. Was there a reason why normative data was based on the data from mechanical Turk instead of from other participant groups? Norm is lower than online samples. Potential issues here if lower norm used in SR recruitment if mechanical Turk not best matched normative group. Could be linked to motivation but arguably applied SRs would be more similar to the most motivated control group.

*We have added justification for the use of Mturk as the normative sample to the manuscript on pg 10 ln 234. Mturk was used to establish the test norm as it is currently the most used sample for online testing, is generally more diverse than students and representative of an online-literate population (e.g. Crump et al., 2013). We also believe the online samples are skewed relative to the norm because they are influenced by the recruitment strategies that targeted super-recognisers and therefore aren’t suitable for deriving the SR classification thresholds (see pg 13 ln 285). As for the second point, there is limited evidence of any effect of motivation on face identification ability (Bobak et al., 2016), therefore we would expect a motivated control group to perform just as well as our normative group on this test. Instead, we attribute the higher mean in online samples to our recruitment strategies using headlines like ‘So, you think you're good at recognizing faces’, which may appeal more to higher performers that have some insight into their own face identification abilities.

3. Line 308. Is there a reason why convergent reliability is based only on lab sample when online sample 2 also completed CFMT and GFMT?

*We have updated the manuscript to be clear on this reason on pg 18 ln 388. Online samples showed ceiling effects that would affect the validity of the correlations. Because lab samples showed a more normal distribution and no ceiling effects, we believe these are more representative of the true convergent reliability.

4. In text report of Fig 6. Suggest that reference to variation in performance at individual level is added – these differences are apparent in the scatterplots, but only group level correlations addressed in text. Acknowledgement of variability in individual level results important especially for applied use of the screening test. Would also be relevant to acknowledge individual spread if this was also found for results reported in Table 2 (through in text comment or addition of scatterplot).

*We have added the suggested acknowledgement to the caption for Fig 6 on pg 16 ln 352. 

“Fig 6. Scatterplots of the correlations between the UNSW Face Test, CFMT+ and GFMT for Online Sample 2. These results show the considerable variability in individual performance across the three tests, demonstrating the importance of repeated testing when establishing super-recognition. They also show that the UNSW Face Test does not suffer from ceiling effects, unlike existing tests and which can aid in the identification of super-recognisers.”

5. Line 312 – UNSW face test more strongly correlated with CFMT. It would be interesting to see the breakdown of correlations for Part 1 and Part 2 of the UNSW face test with GFMT and CFMT+. Did part 1 correlate more strongly with CFMT as these tests are more similar?

*A similar point was raised by Reviewers 1 and 2.We now report this analysis in the manuscript on pg 18 ln 396 and in S2 Appendix. The memory task correlates more with the CFMT+ than the GFMT, while the Sort task correlates equally with both of these tests.

6. Line 338 – explained why the sample was used. I found this sentence really helpful and would suggest that where sensible similar sentences are added for the above results sections as it was sometimes confusing to keep track of which group was used and why others were not included.

*We have added sentences to justify each sample choice throughout the manuscript, for example on pg 10 ln 234 and pg 18 ln 388.

Minor

- Line 27, typo. ‘that’ -> ‘than’

*Corrected

- Line 385. Helpful to add context for why criteria can vary (different operational context etc). Otherwise reader may be unsure about the level of performance that is required to be defined as an SR. A sentence on the lack of clarity over definition of SR may suffice.

* We have clarified this point (pg 22 ln 484-489) 

“Although the accuracy threshold used to define super-recognizers varies across studies, we have shown here that stricter recruitment criteria translate to higher performance in participant groups. So from a pragmatic viewpoint, if the goal is to study the highest performing participants on face identification tasks, adopting stricter criteria on screening tests will yield a greater proportion of high performing participants in follow up confirmatory tests.” 

Reviewer #4 

The abstract would be improved by the following minor changes:

- change "available for free for scientific use" to "available free for scientific use"

- change "an important tool for their initial recruitment" to "an important step in initial recruitment"

- change "before completing confirmatory testing" to “before confirmatory testing”

- change “normative data on the test” to “normative data on our screening test”

- change "stricter selection criteria that" to "stricter selection criteria than"

*Thank you. We have adopted all these wording suggestions.

Other comments:

Page 3. Line 55. “identifying super-recognizers based on self-report alone is unreliable”. Zhou & Jenkins (2020) found that high performers consistently underestimate their performance, both in absolute terms (estimated accuracy) and in relation to others (estimated rank).

Zhou, X., & Jenkins, R. (2020). Dunning–Kruger effects in face perception. Cognition, 203, 104345.

* We have added this reference to this paragraph (pg 3 ln 57).

Page 3. Line 56. Change “This coupled with” to “This constraint, coupled with”

*We have rewritten this sentence to make it clearer (pg 3 ln 58).

Page 4. Lines 64–66. “However, existing standardised tests of face identification ability are unsuitable for online testing. For example, the CFMT and GFMT are carefully calibrated psychometric tests intended to be reliable measures of a person’s ability.” It would be helpful to briefly explain why these properties make the tests unsuitable for online testing. They sound like desirable properties on the face of it.

* We have clarified this point in the manuscript on pg 4 ln 72. Please see our response to Reviewer 1 for more details.

Page 4. Lines 69–75. This argument is not well developed. In particular, the sentence beginning “Further” seems to run against the preceding sentence. The paragraph would benefit from a thoughtful rewrite.

*Thank you for this suggestion. We have rewritten this paragraph to make our argument clearer (now pg 4, from ln 83). To reiterate, this test has been designed specifically to reliably identify super-recognizers in large scale online testing. The intention is not to replace existing tests, but to complement them by providing a screening tool than can identify individuals that are likely to excel on diagnostic tests.

Page 7. Line 136. Change “Participants’ scores” to “A participant’s score”

*We have adopted this wording (pg 8 ln 171).

Page 8. Lines 155–162. Avoid excessive repetition of “participants”. e.g. “Next, they sort”, “resulting in a maximum score of 80”.

*We have adopted this wording (pg 9 ln 195).

Page 10: Lines 214–217. Briefly explain what the percentage values in the main text refer to, and their relation to Table 1.

*These percentages refer to the difference in accuracy between the normative sample and each other sample, along with the statistics of that comparison. We have clarified this in text (now pg 12, from ln 263).

Page 14. Lines 296–298. One possible explanation is that the high task difficulty forced participants to guess on a high proportion of trials. There is no reason that ‘pure’ guesses should be correlated across attempts.

*We have edited this section to acknowledge this on Pg 17 ln 370.

Page 17. Lines 353–355. This is an interesting result. It could be taken to mean that face recognition ability peaks around age 30. However, it could also be taken to mean that it is easier to recognise faces that are (roughly) your own age (i.e. the ‘other-age effect’). I don't think the current data, on their own, can distinguish these two possibilities. It might be worth briefly commenting on the ambiguity, if only for the sake of future systematic reviewers and meta-analysts.

*We have updated the manuscript on pg 21 ln 470 to include this argument. In both the CFMT and UNSW Face Test, the faces were young adults and so it is possible these results reflect an own age bias whereby younger participants benefited from the choice of stimuli. However, because the average age of faces in the UNSW Face Test (19 years old) is almost 10 years younger than the peak performance age, this explanation alone cannot account for this age effect.

Reviewer #5

Major (note that page numbers refer to manuscript pages, not the complete pdf file):

Page 8, line 161: It would be nice if there were not only results for overall accuracy, but also for hits and false alarms.

* We have added hits and false alarm rates as well as Signal Detection theory analysis to the S2 Appendix and have included the response rates in S1 Dataset.

Page 10, line 10: Please add infos on sample size to the legend of Fig. 2.

*We have added the sample size to Fig 2

Page 10, line 214: It is pretty obvious, but it would still make reading easier if it was made clear that the percentages refer to the differences.

*This was also suggested by reviewer 4. We have amended the text “Difference to normative” to clarify this (now pg 12, from ln 263).

Page 14, line 291: Test-retest reliability is not that great, and might be distorted by using the same version within one week. Maybe the authors could avoid potential repetion effects by calculating test-retest reliabilty for (post-hoc) contructed paralell versions?

*Test-retest reliability is within the range of other tests of this ability. Published CFMT data shows test-retest reliability of 0.7 (Wilmer et al., 2010). For the GFMT, unpublished data from our lab shows test-retest reliability over a week of r(86) = .591, p < .001 (Tyler et al., in prep.). As we have discussed in response to Reviewer 4, part of the reason for lower reliability is likely to be the floor effects on this test, which are a product of focussing our attention to the upper-end of the measurement scale (pg 17 ln 370).

Page 15, table 2: Isn’t it surprising that the correlation between UNSW FT Time 2 and CFMT is (numerically) larger than the correlation between UNSW FT Time 1 and UNSW FT Time 2? At the same time, I was surprised that the correlation between UNSW FT and CFMT+, which is an established tool for identifying super-recognizers, is rather low, and lower than for the “normal” CFMT. Do the authors have any explanations for these findings?

*It is our understanding that these correlations are affected by the test-retest reliability of the UNSW Face Test. As Wilmer et al. (2012) notes, “geometric mean of the reliabilities of two measures provides a theoretical upper bound on the correlation that may be obtained between them” (pg. 364). Based on the test-retest reliability, most of these correlations are at or close to this upper limit. As for the lower correlation between the CFMT+ and CFMT, this may be due to the ceiling effects on the CFMT+ in Online Sample 2 lowering the association between the two tests.

Page 20, line 435: Why was it decided to take the normative sample from US residents, when all other samples were from Australia?

*We have added justification for the use of Mturk as the normative sample to the manuscript on pg 10 ln 234. Mturk was used to establish the test norm as it is the most used sample for online testing, and is generally more diverse than students and representative of an online-literate population (e.g. Crump et al., 2013). See our response to Reviewer 3 for more details.

Page 20, line 440: I think the categorization of participants is not precise and in some cases probably wrong: Is a person, whose ancestors were born and have lived for generations out of Africa, automatically “African”? What is meant here is probably the looks. Along these lines, the term “ethnicity” may also be wrong in this context, because it includes a cultural component. Of course, in real life, there is often an association between physical facial appearance and cultural background, but there is no causal relationship. Using the terms race (which is per se a problematic and inprecise term) and ethnicity interchangably, when acutally talking about the physical appearance of a person, means equalizing culture and biology, which seems very problematic to me.

*This point was also raised by the editor and reviewer 1 and we agree. We have improved our descriptions of participant and face ethnicity throughout the manuscript. We note that references to race and/or ethnicity are not judged by us but self-identified in all cases and have noted that in the manuscript on pg 8 ln 157 for the test stimuli and on pg 25 ln 554 for the participants.

Page 21, line 452: I would also pass a similar criticism (i.e. lack of precision) on the term “mixed-race”? How much and what kind of mix has to be there to count as “mixed-race”? In most regions of the world most people are more or less mixed, isn’t it?

* For all of the ethnicity descriptions, the participant self-identified with one the choosen label. This means that those identifying as “mixed” did so of their own accord. We appreciate that psychologically speaking these particular terms may be imprecise, but given they are the participants' self-identified ethnicities, we believe it is best to continue to report them this way. 

Minor:

Page 5, line 102: delete “and” (or add “were” after “and”)

Page 6, line 119: add group or sample after “demographic”

Page 6, line 119: “for” example (rather than “or”)

*Thank you. We have adopted all of these wording suggestions.

Reviewer #6

Major critiques

(1) Lack of quality assurance: The test represents an uncontrolled, online-available test, with insufficient measures implemented for quality assurance to mitigate artificial inflation or reduction of critical thresholds for super-recognizer identification. Two aspects that cannot be identified are repeated participation and data generated by non-human respondents. 

*We have added justification for the use of Mturk as the normative sample to the manuscript on pg 10 ln 234. Mturk was used to establish the test norm as it is the most used sample for online testing, and is generally more diverse than students and representative of an online-literate population (e.g. Crump et al., 2013). We also note that for this sample, we applied strict quality assurance measures to generate unbiased norms. This included assuring that each participant only completed the test once and that each data point was linked to a high quality Mturk profile. We then used these norms to establish consistent super-recognizer thresholds which could be applied to all of the online samples. 

Additionally negligent is the situation that Ss are merely instructed to use a laptop or desktop, but that mobile device usage is neither disabled, nor (apparently) considered in the analysis, and no form of screen calibration is implemented. 

*The UNSW Face Test currently does not work on mobile devices (smartphones and tablets). It can be opened and started by participants, but they cannot respond and proceed in the test. Also, while the quality and settings of participant’s screens or monitors may vary because this is an issue that would affect any online testing and we are assured by research showing consistent replication of lab-based effects in MTurk samples (e.g. Crump et al., 2013). Given it is an online test, there are limits to what is possible – and ultimately it is not possible to standardise the size of an image on a participant’s retina even with explicit instructions.

Further effects that cannot be ascertained relate to partial data exclusion (i.e., what happened to data from Ss who completed only the first task?) and the (unexplained) fixed task order (resulting in any correlations between measures taken beyond the first being potentially contaminated by order effects).

*Fixed task order (like used in the CFMT) produces a more stable measure of individual differences in ability and is preferred over fully randomised test trials (Mollon et al., 2017). For this reason, we are justified in fixing the trial order for the sort task. For the recognition memory task, we chose a randomised order so as to randomly distribute memorial advantages from salient factors like primacy and recency effects across items. 

(2) Test-theoretical limitations: Screening tools require validation on a cohort of interest identified through independent appropriate measures - a prerequisite to ascertain false negative rate (which cannot be addressed via “confirmatory testing via standardized measures and more detailed cognitive testing”). As this is not accomplished here, the authors’ reasoning is unjustified and circular (e.g. “the UNSW Face Test is a valid and reliable test that is uniquely suited to screening for super-recognizers”, p.6, l.107ff.). 

*Our analyses show that the UNSW-FT is effective in screening for participants that score highly on established tests of face matching and memory ability (see Fig 6). In future work, it will be important to verify that established tests more generally are predictive of real-world face identification performance, but this is beyond the scope of this paper. 

To be clear, the authors provide no details on how normative performance was established, only how normative data were obtained. Normative data should be treated as something of a ground truth against which to measure further samples, but it is unclear why this sample ought to be considered “normative”. No ground truth about SRs was ascertained from it, and they were not retested to confirm their SR status, as this study’s aims (creation of a screening tool) would suggest.

We have clarified and extended our justification of using the Mturk sample as our normative reference cohort (pg 10 ln 234). The purpose of this sample was to estimate the M and SD of the test in this reference cohort. 

As for using the normative group to test the accuracy of confirming SR status, this was the motivation of the analysis presented in “The effectiveness of the UNSW Face Test as a screening tool” section, which shows how well these thresholds select top performers on other face identification tests. As shown in Fig 5, the UNSW Face Test is good at predicting high face identification ability as measured by other already well-established psychometric face identification tests.

(1) Stimulus material/selection: The authors acknowledge the importance of considering the ethnic composition of a cohort. In stark contrast, as stimuli were taken from a database “of 236 consenting undergraduate students” (p.6, l.122f.), without information concerning the age composition, we must surmise that the stimulus set suffers from a severely restricted age range. This is problematic for the development of an ecologically valid SR screening tool given same-age performance biases.

*We have updated the manuscript to include the mean age and SD on pg 7 ln 151. To protect the privacy of the faces in the test, we have not made the demographics of the test stimuli publicly available, but it will be supplied to researchers upon request. We have updated the manuscript on pg 21 ln 470 to address the potential age bias in this test. We note that in leading tests of this ability – CFMT and GFMT – the ages of the faces were almost entirely young adults.

Additional concerns

The manuscript appears conceptually fragmented, with sections providing at times inconsistent or

apparently conflicting information, as detailed in the examples (1-3) below.

(1) 3-fold rationale for creation of another online test of face cognition

Referring to the CFMT+ and GFTM the authors state that

a) “super-recognizer7s typically achieve ceiling or near-ceiling accuracy on existing standardised tests” (p.4, l.82f.)

b) “existing standardised tests of face identification ability are unsuitable for online testing” (p.4, l.62ff.)

c) both “use highly standardised images and captured under optimal studio conditions … do not reflect the challenge of real-world face identification” (p.5, l. 101ff.).

Regarding the above aspects, note that the authors

a) state that their test “enables researchers to apply stricter selection criteria that other available tests, which boosts the average accuracy of the individuals selected in subsequent testing.“ (p.2, l.26ff.). They later “propose that researchers and employers verify the super-recognizer status of those who score highly on the UNSW Face Test in controlled conditions using existing standardised tests, such as the CFMT and GFMT.” (p.4, l.73ff.)

This logic is flawed; naturally, Ss identified with a more challenging tool would excel in less sensitive ones.

*The purpose of the using the UNSW Face Test as a screening tool is to predict individuals who would do well on future (validation) tests, as this is currently the best method of confidently identifying a super-recogniser. It is also not a guarantee that individuals who do well on one test will on another, as the predictiveness is determined by the underlying convergence between tests. Therefore, it is vital that the UNSW Face Test does predict future high performance, which we demonstrated in the top row Fig 6, and does so better than the other tests (middle and bottom row of Fig 6).

b) themselves (continue to) use online versions of the CFMT and GFTM to identify

super-recognizers.

*We have revised this paragraph to clarify our position on this (pg 4 ln 72). As we state in the introduction, we believe the most accurate strategy for identifying super-recognisers is to verify the high performance of those who score highly on the UNSW Face Test in controlled conditions using existing standardised tests, such as the CFMT and GFMT. For this reason, we suggest that one use of the UNSW Face Test is that it can help retain exisiting psychometric tests for use as confirmatory tests as opposed to large scale screening.

c) the authors disregard existing tools that specifically uses ambient images in the context of a well-established paradigm: the Models Memory Test (also available as an online version), despite referring to this paper in their manuscript (Bate et al., 2018; reference #25).

*We apologise for omitting this reference when discussing ambient image and have updated that section to include that citation. It was not our intention to suggest that the UNSW Face Test is the first or only test that includes ambient images, but rather that it is a better test for having done so. Had tests like the MMT been published and available when we began data collection for the UNSW Face Test (which was prior to 2018) then we may have considered including it in our test battery but unfortunately it was not an option until data collection was already nearly complete.

(2) lack of conceptual precision

• The authors state that with the two comprising experiments, the UNSW “captures people with a general ability to identify faces, across memory and matching tasks ” (p.5, l. 88f). While the first experiment involves an old/new recognition paradigm, the latter at first glance appears to tap into perceptual skills. However, this is not actually the case. The second experiment is later described as a “Match-to-sample sorting task... combin[ing] immediate face memory, perceptual matching and pile sorting “ (p.7, l.152f.). 

Thus, the UNSW test actually comprises two tests involving recognition memory as measured over different delay periods, and under different conditions.

*As suggested by other reviewers, we now provide a full analysis of the results separated by sub-task and report it on pg 18 ln 396. In summary, the Memory task correlates more with the CFMT+ than the GFMT, implying greater reliance on face memory, while the Sort task correlates equally with both of these tests, implying equal reliance on both matching and memory. 

• Further, throughout the manuscript the authors use the term “face identification ability” to encompass any aspect of face cognition. This is especially important in light of the legal implications of the term “identification” as used to refer to the tasks performed by forensic experts, as White and colleagues have described in their previous work, and have conceptually delineated e.g. in Ramon, Bobak & White (2019).

*We used the terminology of face identification as an umbrella term to encompass both face recognition/memory and face matching. The term face identification is the most intuitive for this purpose and we have used it for consistency with our other publications. We agree there is a need to harmonise terminology in the field, but do not believe this paper is the place to address these broader issues.

(3) Unwarranted claims / selective referencing of the literature

• “individual differences […] generalise from one face identification task to another [...] and represent a domain-specific cognitive skill that is dissociable from [...] visual object processing ability” (p.3, l.36ff.)

There is ample evidence suggesting the contrary to both aspects: work focusing on individual differences has provided evidence of performance across tasks of face cognition that can be unrelated (Bate et al., 2018; Bobak, Dowsett, et al., 2016), as well object processing abilities investigated in over 700 cases in developmental prosopagnosia (Geskin & Behrmann, 2018). Moreover, this stands in direct conflict with other statements in the manuscript: “Further, any single test provides an unreliable indication of face identification ability.” (p.4, l.72); “While these abilities may be dissociable to a limited extent (e.g. [2, 25, 30]), the high correlation between them suggests there is substantial overlap in these two abilities” (p.5, l.94ff.)

* We have noted the missing citations in the manuscript and have added both the Bobak 2016 and the Geskin & Behrmann 2018 reference where appropriate to the manuscript. While there is evidence of a dissociation across face identification tasks, we argue that the variance accounted for by these differences is much smaller than variance accounted for by general face identification ability. For example, in Bate et al. 2018, the correlations between the PMT (a matching task) and the CFMT+ and the PMT and MMT (a memory task) was larger than the correlation between the CFMT+ and the MMT (both memory tasks). Similarly, in this same paper, the PCA shows that hits and CR on these tests load onto the same components, suggesting that most of the variance is explained by the same underlying ability in general face identification rather than dissociable face identification skills. So while there is some dissociation between memory and matching tests noted by Bate and others, there is still a significant overlap in general face identification ability across these tasks. 

• “finding super-recognizers is difficult because they make up just 2-3% of the general population” (p.3, l.49f.); Face identification ability is normally distributed, and people at the very top end – ‘super recognizers’ – demonstrate extraordinary innate abilities” (p.7, l.8f.). As White and colleagues have discussed (Ramon et al., 2019), the actual prevalence is yet to be determined; current estimate vary depending on the cutoffs and criteria applied; it is yet to be determined whether SRs form a special group in itself or are just the extremes of a continuum (cf. Young & Noyes, 2019).

*While we agree with the reviewer that the actual prevalence is unclear, this is also the case with developmental prosopagnosia as both definitions are based on test criteria. But as long as standard deviations on test scores are used as the thresholds for super-recognition, as is the current scientifically accepted standard, there will be a somewhat lawful relationship between thresholds and prevalence as indicated by the range of prevalence stated here. We accept this definition and the estimated prevalence may change in future with more research but are happy to provide a historical record of the current scientific thinking at this time. 

As for the final point, we agree that this remains unknown. But whether there are qualitative or quantitative differences between super-recognisers and the general population is irrelevant for our purposes because, either way, they ultimately sit at the top end of the distribution.

• “small sample sizes limit the statistical power of comparisons between super-recognizers and normative sample” (p.3, l. 53ff.) We argue that large quantities of data collected under conditions where the impact of undesired nuisance variables is unknown (see major flaw (1)) are not the solution to all problems. In cognitive neuroscience and neuropsychology, there is a long tradition of carefully designed studies conducted in small, carefully curated cohorts of rare populations, which draw upon specifically developed statistical approaches.

*See response to major flaw 1. We designed the UNSW Face Test to aid in the curation of super-recognizers as small sample size has been as a limitation of earlier work on this group. Further, by creating this online screening test we will enable new researchers to contribute to this field by helping them find and access a larger cohort of super-recognizers in their local area.

• “These abilities are employed to greater or lesser degrees in different professional tasks that super-recognizers have been recruited to perform. For example, in CCTV surveillance, super-recognizers monitor footage for faces they have committed to memory (e.g. [29]), whereas passport officers match photo-ID to unfamiliar travellers.” (P.5, l.90ff.). 

This creates the two misleading impressions, namely that SRs are (a) specifically recruited (authors reference #29) and widely used; (b) perform only memory tests, vs. passport officers who allegedly only make facial comparisons. (a) This is not the case; rather than psychometric testing employed for personnel selection, individuals already within a given organization have been referred to as SRs based on “anecdotal” peer-evaluation. Note also that the MET does not use SRs for recognition and identification (personal communication). To our knowledge and based on an assessment across international security agencies, only a small fraction is interested in this topic/actively pursues their deployment.

(b) The potential areas of deployment are wide-ranging and specifically include perceptual image comparisons (Ramon & Rjosk, in press).

* We have edited the tone of this section to not give a misleading impression for readers (pg 6 ln 119). While we agree there have been instances where personnel have been selected within an organization, we are also aware of organisations that have used test results to recruit high performers for specialist roles on face identification teams (e.g. White et al., 2015) and we are currently working with organizations seeking to do the same. 

We also thank the reviewer for directing us to this in-press publication. We look forward to reading it when it is available. 

(4) Methods: opaque descriptions, and insufficient explanation/justification of decisions hindering

replication

• Criteria used to select heterogeneous ethnicities for target faces, or how foils were rated as similar to particular targets are unclear.

*We have clarified the selection of target and foil faces as requested by Reviewer 2 (pg 7 ln 152). To summarise, our goal was to select pairs of target and foils that were as similar in appearance to each other as possible as judged by the research team. The selection of non-white faces from the image database was done using this same process, pairing them with another face that appeared from the same ethnic background and was similar in appearance.

• Both experiments contain different numbers of trials, and there is no rationale provided for taking the sum of correct answers across these two tasks. Is there a compelling reason to weight them differently? If so, it should be provided; if not, the metric should be unweighted.

*Our rationale is that using the sum of the correct answers provides an unweighted metric that equally scores each response a participant makes. Because the sorting task contains both short-term memory and perceptual matching, while the memory task has just long-term memory, the summed score (80 sort trial + 40 memory trials) would roughly weight these three skills equally. To do otherwise would be to weight the response to one test question as being more predictive of general face identification ability over another and there is no compelling reason why that would be true either. 

• No compelling rationale for including only the two measures reported here. Neither experiment’s performance on its own was correlated with non-facial tests of the same abilities, and the composite score was shown not to correlate with non-facial tests of these abilities. One would expect quite the opposite if “there is substantial overlap in these two abilities” (p.5, l.96).

*We found that the UNSW Face Test scores are predicted by other face identification tests. This was true for both face memory and face matching (even when separated by sub-task, see S2 Appendix). However, it was not predicted by non-face tests, whether they involve memory or matching. Consequently, the convergent and discriminant validity analysis supports the argument that general face identification ability accounts for more variance than the dissociable skills of face memory and face matching, but also that face identification ability is a domain-specific cognitive skill not related to other perceptual or cognitive abilities (Gignac et al., 2016; Richler et al., 2017; c.f. Geskin et al., 2017).

• No analysis of response times is provided. Thus, any SRs identified by the UNSW face test may simply have been trading off speed for accuracy and vice versa. Particularly in an inventory where ceiling effects are not present (and the test is quite difficult), such trade-offs should be of great concern in validating a screening tool. Moreover, no description of instructions to participants was provided: were they ever explicitly instructed to focus on accuracy to the exclusion of speed?

*We have provided total response time for each sub-task and overall in S1 Dataset. Analysis for associations between accuracy and RT are reported in S2 Appendix and show no significant correlations for any sample for either sub-task or overall (S2 Table 2). There was no instruction on timing for the Memory task, but for the Sort task, there was the following instruction regarding response speed and memory decay: “You can take as long as you like to reach a final decision. However, please note that taking too long may not improve performance, as the task requires matching images to a face in your memory, and this memory will fade over time.”

• Granting that the authors “intentionally did not calibrate the difficulty of the test so that mean accuracy was centred on the midpoint of the scale, as is common practice in standardized psychometric tests” (p.4, l.78ff.), they still do not provide any information about how they did calibrate the test. Since they invite “researchers to create their own versions of the test” (p.6, l.117), this information is absolutely crucial to ensure replicability, as changing the stimulus set would obviously have the potential to alter any existing calibration. Likewise, calibration data for the two tasks contained in the UNSW face test were not provided for the normative cohort (or the subsequent samples).

* We suggest that other researchers may wish to adopt the UNSW Face Test paradigm using images of undergraduates at their own institutions. We have provided additional descriptions of how our targets and foils were selected and the types of images that were used (pg 7 ln 152), and hope that this assists other researchers in replicating our overall approach. We would also gladly assist others in these efforts. While we do not believe it is necessary that researchers calibrate their test to precisely the same M and SD we report in this paper, it is expected that following the method described would result in a similarly challenging test. 

References

Bobak, A. K., Dowsett, A. J., & Bate, S. (2016). Solving the Border Control Problem: Evidence of Enhanced Face Matching in Individuals with Extraordinary Face Recognition Skills. PLoS One, 11(2), e0148148. https://doi.org/10.1371/journal.pone.0148148

Crump, M. J., McDonnell, J. V., & Gureckis, T. M. (2013). Evaluating Amazon's Mechanical Turk as a tool for experimental behavioral research. PLoS One, 8(3), e57410. https://doi.org/10.1371/journal.pone.0057410

Gignac, G. E., Shankaralingam, M., Walker, K., & Kilpatrick, P. (2016). Short-term memory for faces relates to general intelligence moderately. Intelligence, 57, 96-104. https://doi.org/10.1016/j.intell.2016.05.001

Mollon, J. D., Bosten, J. M., Peterzell, D. H., & Webster, M. A. (2017). Individual differences in visual science: What can be learned and what is good experimental practice? Vision Res. https://doi.org/10.1016/j.visres.2017.11.001

Richler, J. J., Wilmer, J. B., & Gauthier, I. (2017). General object recognition is specific: Evidence from novel and familiar objects. Cognition, 166, 42-55. https://doi.org/10.1016/j.cognition.2017.05.019

Tyler, R., Towler, A., & White, D. (in prep.). Let’s face it: Holistic processing does not predict individual differences in face identification. 

White, D., Dunn, J. D., Schmid, A. C., & Kemp, R. I. (2015). Error Rates in Users of Automatic Face Recognition Software. PLoS One, 10(10), e0139827. https://doi.org/10.1371/journal.pone.0139827

Wilmer, J. B., Germine, L., Chabris, C. F., Chatterjee, G., Gerbasi, M., & Nakayama, K. (2012). Capturing specific abilities as a window into human individuality: the example of face recognition. Cogn Neuropsychol, 29(5-6), 360-392. https://doi.org/10.1080/02643294.2012.753433

Wilmer, J. B., Germine, L., Chabris, C. F., Chatterjee, G., Williams, M., Loken, E., Nakayama, K., & Duchaine, B. (2010). Human face recognition ability is specific and highly heritable. Proc Natl Acad Sci U S A, 107(11), 5238-5241. https://doi.org/10.1073/pnas.0913053107

---

## [Decision Letter · Decision Letter 1]

21 Oct 2020

UNSW Face Test: A screening tool for super-recognizers

PONE-D-20-16507R1

Dear Dr. Dunn,

We’re pleased to inform you that your manuscript has been judged scientifically suitable for publication and will be formally accepted for publication once it meets all outstanding technical requirements.

Kind regards,

Peter James Hills, PhD

Academic Editor

PLOS ONE

Additional Editor Comments (optional):

Reviewers' comments:

Reviewer's Responses to Questions

**Comments to the Author**

1. If the authors have adequately addressed your comments raised in a previous round of review and you feel that this manuscript is now acceptable for publication, you may indicate that here to bypass the “Comments to the Author” section, enter your conflict of interest statement in the “Confidential to Editor” section, and submit your "Accept" recommendation.

Reviewer #1: All comments have been addressed

Reviewer #3: All comments have been addressed

Reviewer #4: All comments have been addressed

Reviewer #5: All comments have been addressed

2. Is the manuscript technically sound, and do the data support the conclusions?

Reviewer #1: Yes

Reviewer #3: Yes

Reviewer #4: (No Response)

Reviewer #5: Yes

3. Has the statistical analysis been performed appropriately and rigorously? 

Reviewer #1: Yes

Reviewer #3: Yes

Reviewer #4: (No Response)

Reviewer #5: Yes

4. Have the authors made all data underlying the findings in their manuscript fully available?

Reviewer #1: Yes

Reviewer #3: Yes

Reviewer #4: (No Response)

Reviewer #5: Yes

5. Is the manuscript presented in an intelligible fashion and written in standard English?

Reviewer #1: Yes

Reviewer #3: Yes

Reviewer #4: (No Response)

Reviewer #5: Yes

6. Review Comments to the Author

Reviewer #1: I'm happy with the revisions the authors have made. My only suggestion is that they consider adding information to the abstract that provides some information about how the test works (e.g., unfamiliar faces, match-to-sample sorting, recognition memory).

Reviewer #3: (No Response)

Reviewer #4: (No Response)

Reviewer #5: The authors have addressed all my comments satisfactorily. The only remaining detail is that the wording with respect to the description of the participants should also be adjusted in Figure 1 of the appendix.

7. PLOS authors have the option to publish the peer review history of their article (what does this mean?). If published, this will include your full peer review and any attached files.

Reviewer #1: **Yes: **Brad Duchaine

Reviewer #3: No

Reviewer #4: No

Reviewer #5: No

---

## [Editor Report · Acceptance letter]

26 Oct 2020

PONE-D-20-16507R1 

UNSW Face Test: A screening tool for super-recognizers 

Dear Dr. Dunn:

I'm pleased to inform you that your manuscript has been deemed suitable for publication in PLOS ONE. Congratulations! Your manuscript is now with our production department. 

Kind regards, 

on behalf of

Dr Peter James Hills 

Academic Editor

PLOS ONE